

# A model-data comparison of the Last Glacial Maximum surface temperature changes

Akil Hossain, Xu Zhang, Gerrit Lohmann

Alfred Wegener Institute, Helmholtz Centre for Polar and Marine Research, Bremerhaven, Germany.

**Abstract.** Over the Last Glacial Maximum (LGM, ~21ka BP), the presence of vast Northern Hemisphere ice-sheets caused abrupt changes in surface topography and background climatic state. While the ice-sheet extent is well known, several conflicting ice-sheet topography reconstructions suggest that there is uncertainty in this boundary condition. The terrestrial
and sea surface temperature (SST) of the LGM as simulated with six different Laurentide Ice Sheet (LIS) reconstructions in a fully coupled Earth System Model (COSMOS) have been compared with the subfossil pollen and plant macrofossil based and marine temperature proxies reconstruction. The terrestrial reconstruction shows a similar pattern and in good agreement with model data. The SST proxy dataset comprises a global compilation of planktonic foraminifera, diatoms, radiolarian, dinocyst, alkenones and planktonic foraminifera Mg/Ca-derived SST estimates. Significant mismatches between modeled
and reconstructed SST have been observed. Among the six LIS reconstructions, Tarasov's LIS reconstruction shows the highest correlation with reconstructed terrestrial and SST. In the case of radiolarian, Mg/Ca, diatoms and foraminifera show a positive correlation while dinocyst and alkenones show very low and negative correlation with the model. Dinocyst-based SST records are much warmer than reconstructed by other proxies as well as Pre-industrial (PI) temperature. However, there are large discrepancies between model temperatures and temperature recorded by different proxies. Eight different PMIP3
models also compared with  temperature proxies reconstruction which show mismatches with the the proxy records might be due to misinterpreted and/or biased proxy records. Therefore, it has been speculated that considering different habitat depths and growing seasons of the planktonic organisms used for SST reconstruction could provide a better agreement of proxy data with model results on a regional scale. Moreover, it can reduce model-data misfits. It is found that shifting in the habitat depth and living season can remove parts of the observed model-data mismatches in SST anomalies.


## 1 Introduction

The climate state at the LGM usually serves as a vital test case for numerical climate models because of its distinct climate background (e.g. GHG, CO2, etc [Ahn and Brook, 2008; Clark et al., 2009; Zhang et al., 2013]) from present day, as well as
the abundance of proxy data at that time (Duplessy et al., 1988; Bard et al., 2000; Adkins et al., 2002; Peltier, 2004; Gersonde et al., 2005; Clark et al., 2009; Hesse et al., 2011; Gutjahr and Lippold, 2011). As a result, simulating the LGM is recognized as a core experiment in the Paleoclimate Modeling Intercomparison Projection (PMIP) (Kageyama et al., 2013). One fundamental characteristic of the LGM is the presence of massive continental ice sheets over Eurasia and North America, accounting for the sea level lowstand at that time (Clark et al., 2009). Previous studies proposed that these northern
hemisphere ice sheets, especially the North American Laurentide ice sheet (LIS), are of crucial importance on modulating glacial climate (Zhang et al 2014, Ullman et al., 2014; Gong et al., 2015). However, there is a large spread of reconstructed LIS with fundamental different geometries and a wide range of sea level equivalent from ~48 m to ~70 m during the LGM (Liccardi 1998; Argus and Peltier, 2010; Braconnot et al.,  2012; Tarasov et al., 2012; Lambeck et al., 2014). Therefore, this will lead to a wide spread of simulated LGM climate states of which reliability needs to be evaluated with paleo-
reconstructions to narrow down the uncertainty itself.

Terrestrial, marine and ice-core records are the key archives of past climate changes by providing valuable information of different climate elements (e.g. surface temperatre, etc.). These records can give qualitative inferences about climate (Kohfeld and Harrison, 2000) or can be statistically analyzed to provide climate reconstructions ( Waelbroeck et al., 2009;
Bartlein et al., 2011). The latter is thus a useful archive for a quantitative comparisons with the simulations of model (Braconnot et al., 2012).

There have been several studies focusing on model-data comparisons of the LGM climate evolution to identify and explain model-data mismatches. Previous attempts have shown that models can generally simulate the right sign of climatic change
from the LGM to the PI but with less magnitude relative to the reconstructions (Kageyama et al., 2006; Masson-Delmotte et al., 2006; Braconnot et al., 2007, 2012). Based on surface ocean reconstruction as well as pollen- and plant macrofossil-based reconstruction data, Harrison et al. (2014) showed that models overestimate sea surface cooling (especially in the tropics) (Waelbroeck et al., 2009) and consistently underestimate land cooling (especially in winter) (Bartlein et al., 2011). A recent study by Jonkers and Kučera (2017) analyzed core top stable oxygen isotope (δ¹⁸O) values of different planktonic



foraminifera species. They found that planktonic foraminifera ecology exerts a significant influence on the proxy signal since bloom seasons, as well as habitat depth of planktonic foraminifera vary at different locations.

Over the last few decades, compilation of different proxies (e.g., planktonic foraminifera, dinoflagellate cyst, alkenones, planktonic foraminifera Mg/Ca, diatoms, and radiolaria) is widely used to reconstruct LGM sea surface temperature (Rosell-Melé et al., 1995; Nürnberg et al., 1995; Bard et al., 1997; Herbert et al., 1998; Lubinski et al., 2001; Pflaumann et al., 2003; Rosenthal et al., 2004; Barker et al., 2005; Kucera et al., 2005; de Vernal et al., 2005; Greaves et al., 2008; Waelbroeck et al., 2009). Thereby, two parameters might influence the estimations of LGM SST anomalies: changes in bloom seasons (Rosell-Melé et al., 1995; Sikes et al., 1997; Barker et al., 2005; Haug et al., 2005; Davis and Brewer, 2009; Fraile et al., 2009; Kim et al., 2015), and changes in habitat depth of different species (Nürnberg et al., 1995; Bentaleb et al., 1999; Barker et al., 2005; de Vernal et al., 2006; Kim et al., 2015). Specifically, planktonic foraminifera assemblages in the sediment encompass active communities from different water depths and seasons; Dinocyst assemblages may be a more accurate proxy for the reconstruction of sea-surface conditions in cold environments (de Vernal et al., 2005). Alkenones record a temperature signal which reflects the surrounding water temperature during the lifetime of algae. Generally, species-dependent ecological preferences can influence the recorded information and therefore, the reconstructed temperature is subject to changes in habitat depth and seasonality of the alkenone-producing organisms (Müller et al., 1998; Baumann et al., 2000; Andruleit et al., 2003; Haug et al., 2005). Similarly, Mg/Ca ratios from planktonic foraminifera estimates SST over wide ranges of seasons and water depths (Deuser and Ross, 1989; Mohtadi et al., 2009; Regenberg et al., 2009; Fallet et al., 2010). Therefore, comparison with outputs from climate model will help to understand the recording system itself.

In this study, we have performed simulations with six LIS reconstructions in an atmosphere-ocean fully coupled climate model (COSMOS) to explore the "best-fit" LIS that can force a more consistent pattern with proxies during the LGM. In addition, proxy records are compared with all available model outputs (PMIP3 dataset is also included) to assess the potential ecological effect on data interpretation. This exercise will thus provide us an unprecedented insight of the LGM climate in both data and model world.

## 2 Methods and data

In this study, a comprehensive fully coupled Earth System Model, COSMOS (developed mainly at the Max Planck Institute for Meteorology (MPI), Hamburg), has been used. Our version includes the ECHAM5 atmosphere model (Roeckner et al., 2003), complemented by a land-vegetation model JSBACH (Raddatz et al., 2007), at T31-resolution with 19 vertical layers as well as the ocean model MPI-OM (Marsland et al., 2003) in GR30 resolution with 40 uneven vertical layers. The setup used in this study is identical to the COSMOS-1.2.0 release (Stepanek and Lohmann, 2012) which is used for glacial climates (Gong et al., 2012; Zhang et al., 2013, 2014; Werner et al., 2016).

Two different simulations were performed in the study, one for the pre-industrial (PI) and one for LGM climate. For the PI climate, COSMOS has been run using PI boundary conditions (greenhouse gas concentrations, orbital forcing, ice-sheet topography, land surface and ocean bathymetry) (Wei et al., 2012). For the LGM simulation, greenhouse gas concentrations ($CO_2$ = 185 ppm; $CH_4$ = 350 ppb; $N_2O$ = 200 ppb) and orbital forcing as well as surface boundary conditions (ocean bathymetry, terrestrial topography, runoff routes according to ice sheet reconstruction) are imposed in accordance with the PMIP3 protocol (Braconnot et al., 2012; Zhang et al., 2013). An increased global salinity (1 PSU (Practical Salinity Unit) added compared to modern values) accounts for an LGM sea level drop of approximately 116 m. The present experiments for six ice-sheet reconstructions were run for at least 1000 model years. Therefore, both simulations are rated as equilibrated and considered the last 100 model years for analyses.

Six alternate reconstructions of LIS topography with different elevation have been investigated in this study e.g., the reconstruction of ICE-6G v2.0 (Argus and Peltier, 2010; hereafter Ice6g), ANU (Lambeck et al., 2014; hereafter Lambeck), GLAC-1a (Tarasov and Peltier, 2004; hereafter Tarasov), Licciardi et al. (1998; hereafter Licc) and Gowan et al. (2016; hereafter Gowan) are used. Among them, Ice6g, Lambeck, and Tarasov provide overall higher elevation than the remaining two. Another one is the ice-sheet provided for the PMIP3 LGM experiments which is a blended product of three different ice-sheet reconstructions including Tarasov, Ice6g, and Lambeck by averaging them (hereafter LGMctl). The experimental designs and boundary conditions of the LGM simulations are imposed according to the PMIP3 protocol (Braconnot et al., 2012).

The model results of our study have been compared with the LGM continental temperature and precipitation reconstruction by Bartlein et al. (2011), which is mainly based on subfossil pollen and plant macrofossil data. This dataset includes reconstructions of four temperature variables: mean temperature of the warmest month (MTWA), the growing degree days





above a baseline temperature of 5 °C (GDD5), mean temperature of the coldest month (MTCO) and mean annual temperature (MAT) (Bartlein et al., 2011). The dataset considers a quantified estimate of combined uncertainties arising from the age scale uncertainties, data resolution and sampling, calibration model uncertainty, and analytical uncertainties. The
original site-based reconstructions were re-gridded as anomalies (LGM-PI) on a regular latitude/longitude grid of 2° by 2° to facilitate comparison with climate model outputs. The grid-cell anomaly value was obtained by averaging, while grid-cell uncertainty is a pooled estimate of the standard error (Bartlein et al., 2011).

The Multiproxy Approach for the Reconstruction of the Glacial Ocean Surface (MARGO) project in 2009 has compiled and
analyzed an updated synthesis of seasonal sea surface temperatures (SSTs) and seasonal sea ice cover during the LGM (Kucera et al., 2005) based on all prevalent microfossil-based (planktonic foraminifera, diatoms, dinoflagellates and radiolarian abundances) and geochemical (alkenones and planktonic foraminifera Mg/Ca) palaeothermometers from deep-sea sediments (Waelbroeck et al., 2009). Different types of records provide various information about ocean surface conditions: for example, alkenone data only give a measure of mean annual SST while foraminiferal assemblages can be
analyzed statistically to obtain seasonal variation in SSTs (Waelbroeck et al., 2009). The various datasets have been subsequently combined to provide gridded datasets of annual SST (SSTann), winter SST (SSTwin), summer SST (SSTsum), where winter is defined as January, February, March in the northern hemisphere and July, August, September in the southern hemisphere (and vice versa for summer). The MARGO dataset combines 696 individual SST reconstructions. The coverage is especially dense in the tropics, the North Atlantic and the Southern Ocean while several oceanic regions remain
undersampled: for example, the subtropical gyres, especially in the Pacific Ocean. The entire SST dataset was projected onto a regular grid of 5°×5° resolution by averaging individual site-based reconstructions that fall into the same cell, weighted by an index of the reliability of each contributing reconstruction (Waelbroeck et al., 2009). The representativeness of an LGM SST estimate at a given site depends on the number of samples per core (a larger number increases the representativeness) and on the quality of the age model for each core (Waelbroeck et al., 2009).

To check whether other models also show the same pattern as ours, the proxy-derived observational data has been compared with different PMIP3 model experiments (listed in Table 1) as described above for the LIS simulations.

## 3 Results

### 3.1 Data Model Comparison: Sea surface temperature changes

Due to the prescribed change in glacial ice-sheet configuration, greenhouse gas concentrations, and orbital parameters, the simulated LGMctl run annual mean SST is 3.1 °C colder than the modelled PI climate in ice-free ocean areas. Most of the
regions show a uniform cooling in the range of −2 to −4 °C (Fig. 1). In the high latitudes of the Southern Hemisphere, the model simulates a pronounced annual mean cooling of SST (up to −2 °C) around Antarctica (Fig. 1), in line with proxy data (Gersonde et al., 2005). A robust meridional temperature gradient is well simulated close to 40∼45° N in the Northern Hemisphere, and the most pronounced cooling is found off (adjacent to Greenland) in the northern North Atlantic, the eastern coast of Iceland to the eastern part of Nordic Sea, where temperature is decreased by −15 °C (Fig. 1). Both features
agree with reconstructions (Kucera et al., 2005; de Vernal et al., 2006). The result from this study are also in agreement with the temperature change documented by CLIMAP, which suggests that the most robust annual mean cooling (up to −10 °C) observed in the North Atlantic and extended to the western Mediterranean (−6 °C) (Bard and Sonzogni, 1997; Pflaumann et al., 2003; Rosell-Melé et al., 2004).

The reconstructed SSTs from the MARGO project are used to compare with the model outputs and to assess the capability of the current models for simulating SSTs during the LGM. Among 275 temperature data points of MARGO project, the mean LGM SST change in the model run is more than 2 °C warmer (colder) than the reconstructed temperature change in about 24 cases (107 cases), while the model-data variations of LGM anomalies range between −15 and +4 °C (Fig. 1). Several sites with opposite sign (where model showing cooling and observational data showing warming) are located in the northern
North Atlantic and western part of North Pacific. The highest correlation coefficient is found for the Tarasov_LIS ice sheet reconstruction (measured R = 0.16, RMSE = 3.5‰) and the lowest for the Gowan_NAIS (R = 0.10, RMSE = 3.1‰), although correlation coefficient is quite low and almost similar for remaining LIS reconstructions (Table 1). Discrepancies between these model LGM simulations and MARGO project data may result from the seasonal and depth bias which will be elaborated later.

The glacial ocean state has been under debate since the first reconstruction of the LGM sea surface temperatures and sea ice coverage by the CLIMAP project (CLIMAP Project Members, 1976). The SST reconstruction by the MARGO project (Waelbroeck et al., 2009) compared to CLIMAP indicates a more pronounced cooling in the eastern mid-latitude of the



North Atlantic than in the western basin, a 1-3 °C cooling in the western Pacific warm pool (Fig. 1), as well as ice-free conditions in the Nordic Sea during glacial summer. According to MARGO study, in all the ocean basins, there is a large longitudinal gradient in LGM SST anomalies which are absent in the most of atmosphere-ocean coupled simulations of the PMIP2 project (Waelbroeck et al., 2009). A rather uniform SST cooling during the LGM in the range of 2-4 °C has been found (Fig. 1).

### 3.2 Data Model Comparison: Land Surface temperature changes

The annual mean SAT of reconstructed LGMctl run is 5.9 °C colder than the modelled PI climate. Most regions show a rather uniform cooling for all of the model runs in the range of −4 to −8 °C (Fig. S2). Alaska is the only region that shows warmer than average in the model because of the increased distance to sea ice covered Arctic Ocean regions during the LGM, possibly due to the glacial sea level drop of approximately 120 m (Werner et al., 2016). The cold regions are mostly adjacent to the FIS and LIS, e.g., most of central North America and central Europe. There is another region of exceptional cooling located in northern Siberia where the temperature decreased down to −15 °C. The results agree with the temperature change of ensemble-mean LGM by the fully coupled climate simulations within the CMIP5/PMIP3 and PMIP2 projects (Braconnot et al., 2007; Harrison et al., 2014).

For a comparison with proxy data, the model results have been compared with the LGM continental temperature reconstruction by Bartlein et al. (2011), which is mainly based on plant macrofossil and subfossil pollen data. The highest correlation coefficient and lowest deviations are found for the Tarasov_LIS ice-sheet reconstruction (R = 0.41, RMSE = 5.0‰) and the lowest correlation coefficient and largest deviations for the Gowan_NAIS (R = 0.29, RMSE = 5.4‰) (Fig. S2). Different core locations with the largest model-data variations are located near the boundary of the FIS and LIS. These deviations might simply be due to the coarse model resolution of 3.8°×3.8° that cannot resolve small-scale temperature changes close to the glacier area in sufficient detail. Overall, the model results agree well with the reconstructed LGM-PI temperature changes at the different core points (Fig. S2).

### 3.3 Six different proxies SST Annual Mean

As the pattern of SST has shown the similar pattern in all LIS reconstructions and low correlation with MARGO dataset, it has been compared with six individual MARGO proxies. All the proxies agree with the model in the LGM cooling over the eastern part of North Atlantic, Mediterranean Sea, Southern Hemisphere and around Australia. This simulated pattern is recorded consistently by foraminiferal assemblages and few available Mg/Ca estimates (Barker et al., 2005). Most of Mg/Ca, diatoms and radiolaria are in good agreement with the model.

A positive correlation coefficient has been found for four proxies with LGMctl run LGM-PI anomaly, e.g., foraminifera (R = 0.43), Mg/Ca (R = 0.51), diatoms (R = 0.63) and radiolaria (R = 0.82) while the annual mean SST of dinoflagellates (R = - 0.23) and $U^k_{37}$ (R = - 0.04) show negative and very low correlation coefficient. However, large deviations between model LGM SST and temperature recorded by different microfossil proxies still exist (Fig. 3). These deviations are smaller in the Pacific and the Indian Ocean than in the Atlantic. In case of foraminifera, several sites in the Arctic Ocean, east of Greenland, in the South Atlantic around 15-25° S and in the Eastern Pacific around 0-35° S show warming trends where the model shows cooling anomalies. There are several core locations for dinoflagellates in the North Atlantic, e.g., the south and east of Greenland as well as south of Svalbard, are showing warming trends. These deviations in the North Atlantic might be caused by dinoflagellates transfer functions (Waelbroeck et al., 2009).

However, in the most densely sampled Nordic Sea, the reconstructed LGM SST have large uncertainties. In the North Atlantic, there are several data points around 50-60° N warmer than the model. In contrast to the North Atlantic, the SST in Northwest Pacific is warmer at the LGM than present in comparison to the model. These results are mainly based on alkenone unsaturation ratios ($U^k_{37}$) and, rather than representing a warm anomaly in the annual mean SST values, these core points might indicate that the seasonality of alkenone production has changed through time (Minoshima et al., 2007).

### 3.4 Seasonality of the recorder system

A comparison of SST trends of four proxies (for diatoms and radiolaria, we have only Southern Hemisphere local summer data) with local winter (DJF), local spring (MAM), local summer (JJA), local autumn (SON), and annual mean SST, as simulated by the anomalies (LGM-PI) of LGMctl run, indicates which season shows the best agreement between proxy reconstruction and model (Fig. 4a). In the North Atlantic Ocean, the best agreement of planktonic foraminifera, dinoflagellates, and alkenones is found for local summer while Mg/Ca ratio shows the best agreement for local winter (Fig.

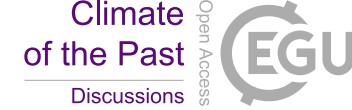



4a). In the rest of the ocean areas, clear evidence for a preferred season is absent. Some cores in proximity to each other show best-fit with different seasons in the model. All these proxies are mainly produced in the spring-summer months as there is very small biological production during the cold and dark winter, so one might expect all of the proxies to record similar signal. However, the individual reconstructions are very different from each other (Fig. 4a).

230

According to Ternois et al. (1996), seasonal variability in alkenones biological production should be considered if they are used as a proxy to reconstruct the temperature. There is a possibility that the SST reconstruction based on alkenones might be biased towards warmer than average climatic conditions or might represent a summer signal prominently if the growth season of alkenone-producing organisms shifted towards the summer (de Vernal et al., 2006). On the other hand, the timing of the maximum foraminiferal production during the LGM did not occur at the same time of the year as present day. The change in the timing of the maximum production of planktonic foraminifera could lead to a bias in reconstructed paleotemperature if the seasonality change is not taken into account (Fraile et al., 2009). Due to the temperature sensitivity of the considered foraminiferal species, during the LGM the most significant production occurred during warmer season of the year (Fraile et al., 2009). Finally, Mg/Ca measurements in surface dwelling foraminifera from the central North Atlantic tend to represent slightly colder than PI conditions in the corresponding water layers (de Vernal et al., 2006).

The correlation between the foraminifera proxy record and the climate simulation of the model is higher for annual mean (R = 0.43) than for local winter (R = 0.10) or summer (R = 0.34). For Mg/Ca, there is a higher correlation for the summer mean (R = 0.57) than for the winter mean (R = 0.47) or the annual mean (R = 0.51). The agreement between the simulated and the reconstructed LGM-PI SST anomalies is still stronger for the local summer than for the other seasonal or annual mean in the North Atlantic Ocean, because the simulated cooling anomaly is much more pronounced for summer than for the annual mean (Fig. 4b). Table 2 shows the correlations coefficient between modelled and reconstructed trends. The correlation between different proxy (foraminifera, dinoflagellates, Mg/Ca ratios and alkenone) records and the model have been increased significantly for best-fit season. For Mg/Ca, there is a higher correlation for the best-fit season (R = 0.59) than the annual mean (R = 0.51) (Table 1 & 2). Foraminifera also show significantly high correlation for best-fit season (R = 0.54) which is even higher than the annual mean (R = 0.43). Dinoflagellates and alkenones show positive correlation for the best-fit season (dinoflagellates, R = 0.01; alkenones, R = 0.19) while they show negative correlation for annual mean (Table 1 & 2).

### 3.5 Habitat depth of the recorder system

In this study, the observational data of MARGO project is composed of different planktonic organisms which are known to be able to shift in the different water columns (Conte et al., 2006). To observe whether deeper layers in the model would be in better agreement with the temperature reconstruction than the surface, the model for different layers of the upper 183 m of the ocean was compared to the proxy records (Fig. 5a). Layers below these depths can be ignored since alkenone-producing organisms require sunlight for photosynthesis.

From Fig. 3, it is observed that most of the core sites in the North Atlantic are in best agreement with the surface layers (0 to 37 m) and the Southern Hemisphere with the subsurface layers (between 70 to 183 m). For dinoflagellates, most of the core points also agree with the subsurface layer. For the Mg/Ca ratio, the same upper layers are also considered, but from Fig. 5a it can be seen that a big part of the record agrees best with the upper level of the ocean although a few cores in the North Atlantic agrees with subsurface layers. For alkenones, a big part of the records agrees best with the deeper layers of the ocean, between 70 and 183 m (Fig. 5a). Rest of the cores have no clear reference to a preferred ocean layer (Fig. 5a).

Investigations of foraminiferal Mg/Ca distributions in Atlantic, Pacific, and Indian Ocean have found that 75–100% of foraminifera species live between 0 to 40 m water depths (van Donk, 1977; Kim et al., 2015). Alkenone-based temperature records do not always precisely represent the SST, because of ambiguities due to the habitat depth of alkenone-producing organisms (Ohkouchi et al., 1999; Lee and Schneider, 2005). The planktonic foraminifera live over a range of depths, mostly between about 75 and 250 m water depth (de Vernal et al., 2006).

The correlation between the reconstructed and simulated LGM-PI anomalies has also been increased significantly for best-fit depth. A higher correlation has been found for the best-fit depth mean (R = 0.65) than for the annual mean (R = 0.51) in case of Mg/Ca (Table 1 & 3). Foraminifera also show significantly high correlation for best-fit depth mean (R = 0.54) which is even higher than for annual mean (R = 0.43). A positive correlation has been found for the best-fit depth in case of dinoflagellates and alkenones (dinoflagellates, R = 0.16; alkenones, R = 0.36) (Table 3) while they show negative correlation for annual mean (Table 1).



If we consider best-fit seasons and depths together at the same time at a specific data point in the model simulations, the mismatch between the climate simulation of the model and different proxy records has been decreased, while correlation coefficient has been increased significantly (Fig. S4; Table S5). Planktonic foraminifera inhabit a wide range of the water column and even show variability in their seasonal abundance. The average proxy signal in sedimentary foraminifera is weighted towards conditions at the season and depth of calcification rather than reflecting annual mean surface conditions (Jonkers and Kučera, 2017).

### 3.6 Data Model Comparison: PMIP3 models

In most of the PMIP3 models, tropical cooling is more pronounced than in the MARGO reconstruction. The highest mismatch between the data and model is located in the Northwestern Pacific. All of the models produced a significant cooling of 4-6 °C during LGM in the Northwestern Pacific, whereas a few MARGO records indicated opposite result (warming of 2 °C or higher). Thus, the large discrepancy between data and model is likely caused by the large uncertainties in the reconstructed data. The models and MARGO both show a more uniform LGM cooling in the Indian Ocean than in Pacific and Atlantic (Fig. 2) (Wang et al., 2013).

In this study, we analyze simulations from the PMIP3 model experiment to test the capability of the current model for simulating the LGM sea surface temperatures with particular attention on model-data comparisons. Therefore, the anomaly of the LGM and PI simulated SST fields of all PMIP3 models have been compared with MARGO data-set and also with four individual proxy-based SSTs separately. A large mismatch and low correlation are found for most of the cases (listed in Table S2). Overall, the anomaly of the LGM and PI SST fields simulated by the PMIP3 models and the LIS simulation runs are comparable. Because of space limitations, all individual model anomalies and their agreement/disagreement with the proxy-derived SST trends is not shown here (shown in Figs. S5-12). Instead, the ensemble median (Fig. 2a) typically displays the common signal. In this case, it is the mean value of the fourth and fifth ensemble member out of eight models which are ordered according to ranked values. However, among all models, IPSL-CM5A-LR shows the highest correlation and lowest RMSE with the MARGO data-set (Fig. 2b; Table S2). Since the results of the PMIP3 runs show large mismatches, we have continued to the next step. The habitat depth, seasonality, and habitat depth-seasonality together in all models have been compared with individual proxies (listed in Table S3-4 and S6). In this case, also correlation increased significantly. Overall, the agreement between the PMIP3 models and the SST reconstructions is almost similar to the case of COSMOS.

## 4 Discussions

### 4.1 Uncertainties of the SST reconstructions

From the above data-model comparison, it is observed that several cores in the North Atlantic show opposite sign that means where the model shows warming, proxies show cooling. For this reason, it is assumed that SST takes into account seasonal and depth biases. The alkenones and dinoflagellates reconstructed temperature trends show a negative correlation to the simulated pattern. The observed mismatches between the model simulations and the proxy records might be caused by misinterpreted and/or biased proxy records as well as by model deficiencies.

Uncertainties in the interpretation of sea surface temperatures estimate based on alkenone-derived mainly stem from the unknown water depth and season of alkenone production as well as from the lateral advection of alkenones (Glacial Ocean Atlas, 2017). In general, marine planktonic diatoms are particularly abundant in nutrient-rich waters with high biomass, and during seasonal blooms in the open oceans (Cervato and Burckle, 2003; Bopp et al., 2005; Armbrust, 2009; Malviya et al., 2016). Investigations on sediment traps reveal that the signal recorded in the sediment record is primarily generated during summer and early fall. That's why it has been argued whether winter temperatures can be reconstructed from radiolarian assemblages. Foraminiferal Mg/Ca is believed to be influenced by pH, salinity, and dissolution (Glacial Ocean Atlas, 2017). Uncertainties in the calibration of Mg/Ca for temperature are large in the low-temperature domain (Lea et al., 1999).

### 4.2 Seasonal biases

The interpretation of our data-model comparison suggests Mg/Ca are winter biased, while foraminifera, dinoflagellates, and alkenones are summer biased. One source of uncertainty in dinocyst is the low productivity and fluxes, particularly in the Nordic Sea, which could have resulted in over representation of transported material (de Vernal et al., 2005). The seasonal contrast of temperature or annual amplitude of temperature is a source of uncertainty for planktonic foraminifera. The seasonality of the temperature signal depends on thermal diffusion and stratification in the upper water layer. In the open ocean, particularly in modern offshore of the North Atlantic, the weak stratification advances high thermal inertia in a thick





mixed layer, which creates low thermal amplitude from winter to summer. Because of this, most open ocean proxies commonly give a mixed temperature signal which does not allow seasonal temperatures to be easily differentiated (de Vernal et al., 2006).

In the present-day ocean, calibrations of alkenone based on mean annual SST appear to result in smaller errors than those
based on monthly or seasonal temperature (Müller et al., 1998). This suggests a production during months or a continuous production during the year when SST values are correlated to the annual mean. Records of alkenone-based reconstructions of SSTs have been analyzed accounting for shifts in the seasonality of alkenone production (Haug et al., 2005). Therefore, in the North Atlantic, alkenone production might be more concentrated in summer months during the LGM than at present, which would impinge in the analysis of the outcomes and perhaps in the calibration. In the high-latitude, the time of
maximum production of alkenone could conceivably occur during the summer, rather than during the autumn or spring (Antoine et al., 1996; de Vernal et al., 2006). The degree of seasonal bias might be dependent spatially since the biogeographical characteristics of the ocean differ from one place to another (Prahl et al., 2010). As summarized by Lorenz et al. (2006), the maximum production of coccolithophorids occurs in summer in high latitudes (Baumann et al., 1997, 2000), which agrees with the idea that $U^K_{37}$ record summer temperature signal (Sikes et al., 1997; Prahl et al., 2010).
Satellite data also agrees with the idea of summer-biased alkenone records (Iglesias-Rodriguez et al., 2002). Seasonality in phytoplankton production is commonly less pronounced in tropical and subtropical regions (Jickells et al., 1996), and alkenone-derived SST from low-latitude sites are therefore more likely to be representative for temperatures close to the annual mean values (Müller and Fischer, 2001; Kienast et al., 2012).

**4.3 Habitat depth biases**

Changes in the habitat depth of the SST over the LGM result in deviations between model simulations and proxy records. Such changes in recording season and habitat depth could have been caused by changes in insolation over the LGM or by related changes in the nutrient distribution and ocean temperature that the individual proxies are exposed to (Lohmann et al.,
2013). Comparing the reconstructed LGM temperature trends at model levels in the upper 183 m does not remove the discrepancy between models and proxies. Planktonic foraminifera live over a range of ocean depths, mostly between about 75 and 250 m water depth (de Vernal et al., 2006). A recent study (Kim et al., 2015) suggested that foraminiferal Mg/Ca record temperatures at depths of 0-40 m in the water column. Concentration of $U^k_{37}$ is high in the subsurface water column of the central Pacific Ocean (Lee and Schneider, 2005) agree with model simulation. Most of the planktonic foraminifera
species in polar waters inhabit subsurface waters (Bé and Tolderlund, 1971) and may be found below the thermocline as it is frequently the case in the Arctic, subarctic and other stratified waters where it occurs along or below the halocline (Kohfeld et al., 1996; de Vernal et al., 2002; Hillaire-Marcel et al., 2004; de Vernal et al., 2006). Planktonic foraminifera normally migrate vertically throughout their life cycle, forming calcite at deeper layers as they mature. This temperature/depth migration results in heterogeneity of Mg/Ca ratios within tests of individual foraminifera (Nürnberg, 1995; Jha and
Elderfield, 2000; Elderfield and Ganssen, 2000; Benway et al., 2003). It has been observed that various planktonic species add a calcite crust in colder, deeper water immediately prior to reproduction (Barker et al., 2005). Many species of planktonic foraminifera live at depths higher than 50 m (Erez and Honjo, 1981; Deuser and Ross, 1989; Anand et al., 2003) which agrees with the studied model simulation.

The results from the habitat depth and seasonality are based on the model output which does not provide any diagnostic on the planktonic organisms real ecological behavior. However, they provide an oceanic regions mapping where even small changes in the ecology of planktonic organisms can have huge consequences on the reconstructed SST anomalies. It reinforces the idea that proxy organisms may be affected by ecological specificities (Leduc et al., 2010, Lohmann et al., 2013).


**4.4 Seasonal and habitat depth biases**

As we see from above observation that considering blooming season or habitat depth of the proxies cannot resolve the model data mismatch completely, we consider shifts in depth habitat and growing season at the same time from the LGM to PI
which reduces the model-data disagreement (Fig. S4; Table S5 & S6).

It is questionable whether proxy-recording organisms act in such a way, as they would likely try to hold their preferred ecological conditions by changing their blooming seasons in a way which mitigates the climate changes (Mix, 1987). Fraile (2008) and Fraile et al. (2009) using a planktonic foraminifera model analyzing the seasonality of the foraminifera showed
that the organisms usually record a weaker temperature signal when the global temperature change is applied. By decreasing the global temperature by 2 ºC and 6 ºC, they did a model sensitivity study and observed a shift in abundance of the





maximum planktonic foraminifera towards warmer seasons, which would reduce the temperature trend recorded in Mg/Ca (Fraile et al., 2009).

On the contrary, planktonic organisms have several limiting factors such as temperature, nutrient, and light-availability. When those factors alter oppositely, the organisms try to change their living season without modifying their basic ecological requirements. For example, nutrient or food availability might shift towards autumn or spring so that living season might change accordingly. To explain such changes, more research using complex ecosystem models of different planktonic organisms need to be performed, such as ecophysiological models, used to reproduce the growth of planktonic foraminifera (Lombard et al., 2011).

For our model-data comparison, it is worth to mention that climate models have some limitations and are unable to represent the full complexity of the physical Earth System. The proxy records used in most of the studies are more often located in coastal areas, and climate models do not well represent these regions because of their low resolution (Lohmann et al., 2013). Coastal areas may be particularly sensitive to external forcing, as their thermal inertia is lower than the open ocean due to land-ocean interactions and a shallower thermocline. Moreover, the representation of mixed layer dynamics may be essential to improve climate simulations and its agreement with palaeoceanographic reconstructions.

## 5 Conclusions

We evaluated the ability of different Earth System Models to capture marine and terristerial temperatures for the LGM. For our model simulations, we considered different versions of COSMOS (while changing the LIS) and models participating in the PMIP3. The model results have been compared with terrestrial temperature reconstruction data showing a similar pattern and good agreement. SST reconstruction shows deviations with the model results. Several cores in the North Atlantic show opposite signs where model show cooling and proxies show warming trends. It is assumed that SST takes into account seasonal and depth biased.

The amplitudes of the simulated anomalies are significantly smaller than the reconstructed temperature trends by alkenones. This deviation persists for all considered models, even if we take into account seasonality and different water depths at which the recording organisms may have lived. As for the Holocene (Lohmann et al., 2013), this raises important questions as to whether climate models have fundamental deficiencies, and/or whether our understanding of the proxy records still needs to be refined. For the LGM, we find the best agreement between reconstruction and annual mean temperatures in low latitudes. We have found that Mg/Ca show winter biased, while foraminifera, dinoflagellates, and alkenones are summer biased. Comparing the reconstructed LGM temperature anomalies with the model levels of the upper 183 m does not remove the discrepancy between models and proxies. For Mg/Ca ratios, a large number of records fit best with the model surface layer. In case of the remaining three proxies, the best-fit agreement is found in the subsurface layer. Thus, we find mostly the highest agreement between proxy-record and simulated temperature anomalies in the subsurface layer depth. Observational data again compared with different PMIP3 models and found the almost similar pattern as in COSMOS. There may be several mechanisms that can be responsible for the observed mismatch between the reconstructed and the modelled magnitude of the LGM SST anomalies. It is therefore conceivable that the observed mismatch between modelled and reconstructed LGM climate evolution is related to the lack of representativeness of long-term temperature anomalies in climate models as well as to the interpretation of the paleoclimate data. As a logical next step, a direct simulation of the recorder system, as it is now routinely done in the case of $\delta^{18}O$ (e.g., Werner et al., 2016), is required.

**Acknowledgements**. This study is supported by Helmholtz Programme PACES program of the AWI and the BMBF funded project PalMod. We are grateful for constructive comments provided by Dr. Christoph Voelker. We thank the many contributors making their proxy SST data available in the MARGO project. We acknowledge the World Climate Research Programme's Working Group on Coupled Modelling, which is responsible for CMIP, and we thank the climate modelling groups of PMIP3/CMIP5 (listed in this paper) for producing and making available their model output. The US Department of Energy's Program for Climate Model Diagnosis and Intercomparison provides coordinating support and led development of software infrastructure in partnership with the Global Organization for Earth System Science Portals.

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








**Figures**

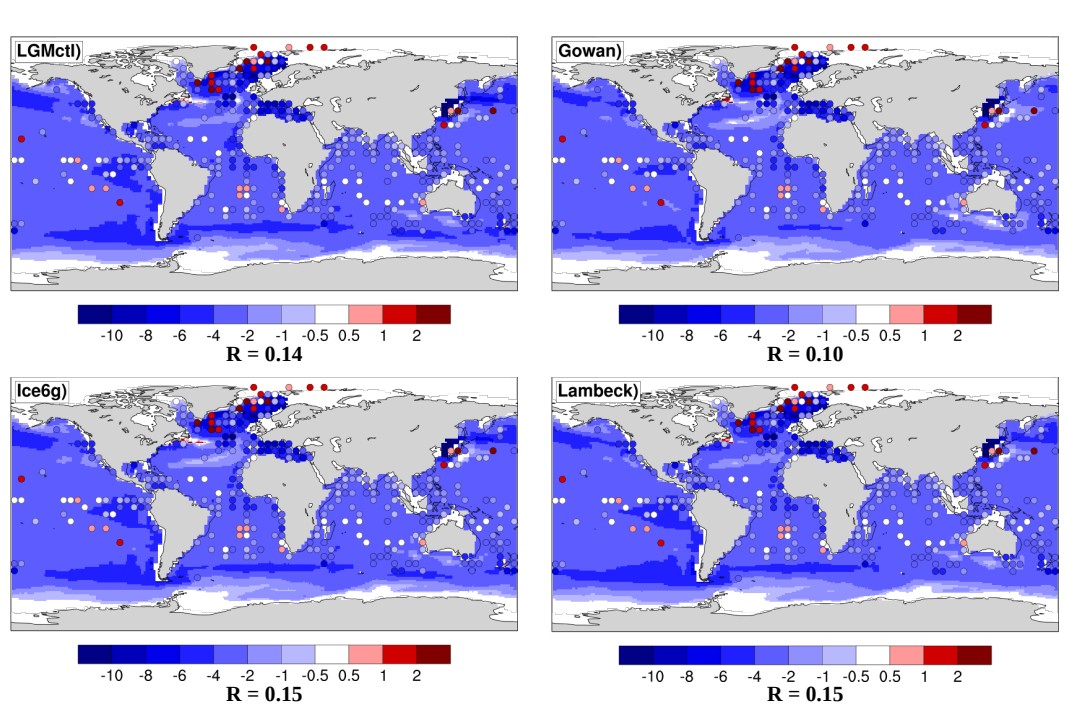



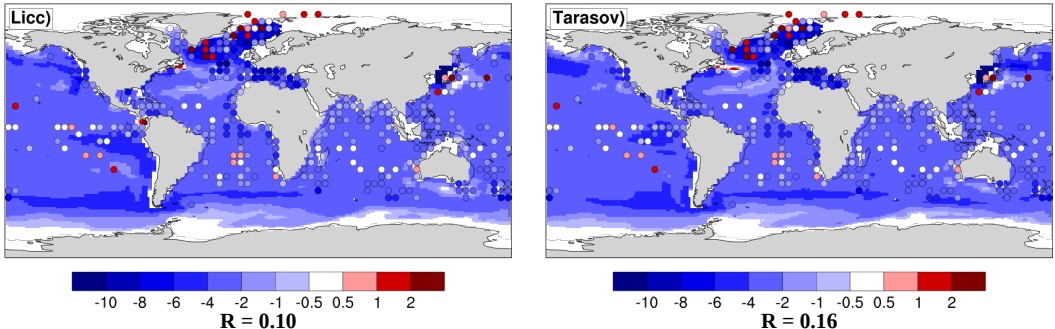

**Figure 1:** Global SST anomalies of the annual mean from six different LIS experiments output compared with MARGO
data-set. Background color fill: simulated global pattern of annual mean sea surface temperature changes between the LGM
and PI climate. The colors fill of the circles show the temperature anomalies as recorded by different proxy records,
respectively.


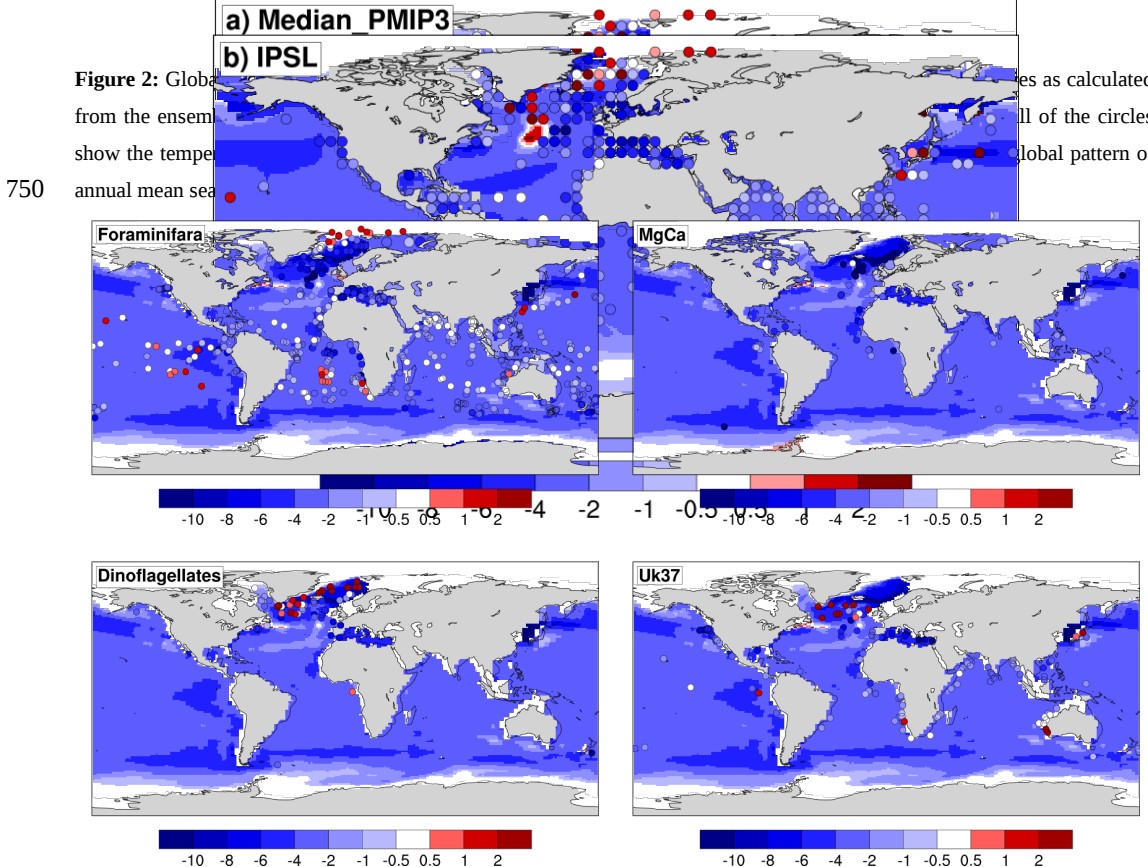

**Figure 2:** Glob[...]es as calculated
from the ensem[...]ll of the circles
show the tempe[...]lobal pattern of
annual mean sea[...]



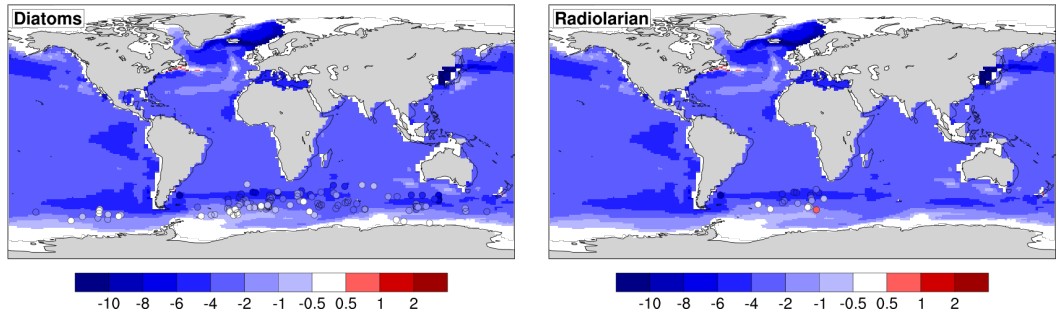

**Figure 3:** Global SST anomalies of the annual mean LGM LIS experiment output, and temperature trends based on planktonic foraminifera, Mg/Ca, dinocyst, alkenones ($U^k_{37}$), diatoms and radiolarian reconstructions of MARGO project. Background color fill: simulated annual mean SST changes between the Tarasov_LIS and PI. The circles localize the three different proxy records, respectively.


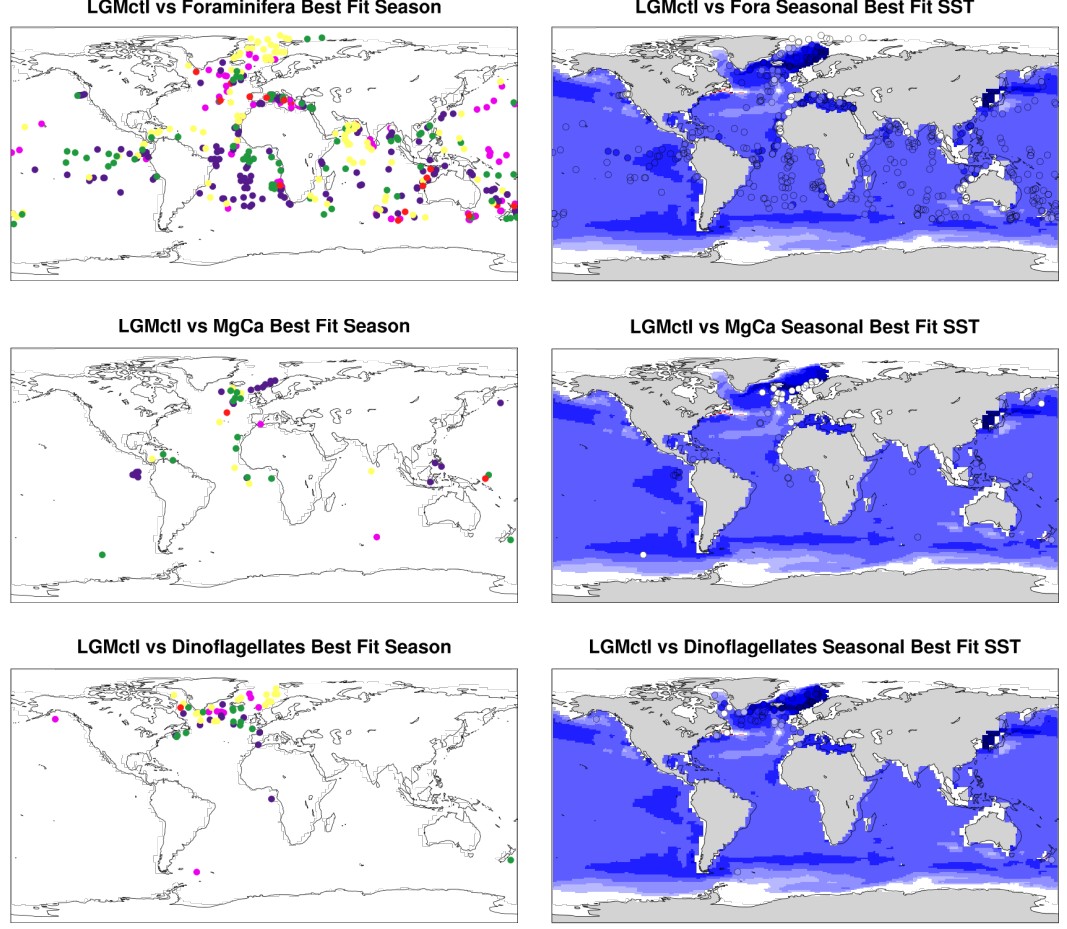



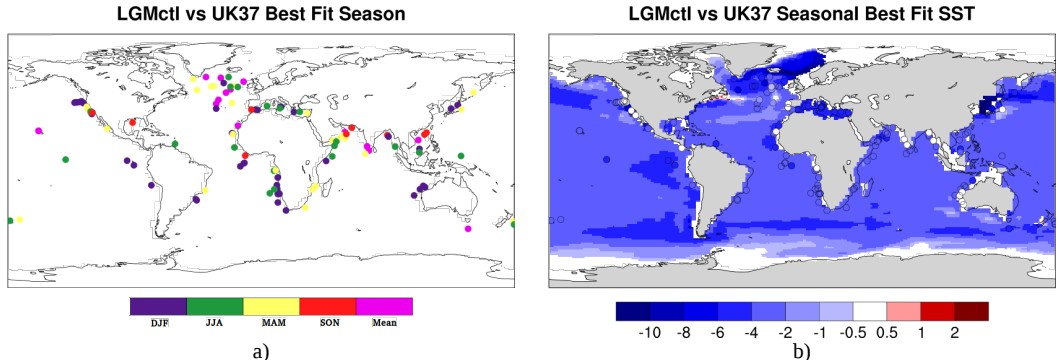

**Figure 4: (a)** The circles localize the foraminifera, MgCa, dinoflagellates and $U^k_{37}$ records and the colors fill of the circles represent the seasonal/annual mean in which the reconstruction agrees best with model. **(b)** Background color fill: simulated global pattern of annual mean sea surface temperature changes between the LGM and PI climate. Colors fill of the circles show the tempera- ture trend recorded by corresponding seasonal/annual mean shown in (a) at the sample locations.



a)                                                                    b)

**Figure 5:** (a) The circles localize the foraminifera, MgCa, dinoflagellates and $U^k_{37}$ records and the colors fill of the circles represent the different depth (m) mean in which the reconstruction agrees best with model. (b) Background color fill: simulated global pattern of annual mean sea surface temperature changes between the LGM and PI climate. Colors fill of the
circles show the temperature change recorded by corresponding depth mean shown in (a) at the sample locations.

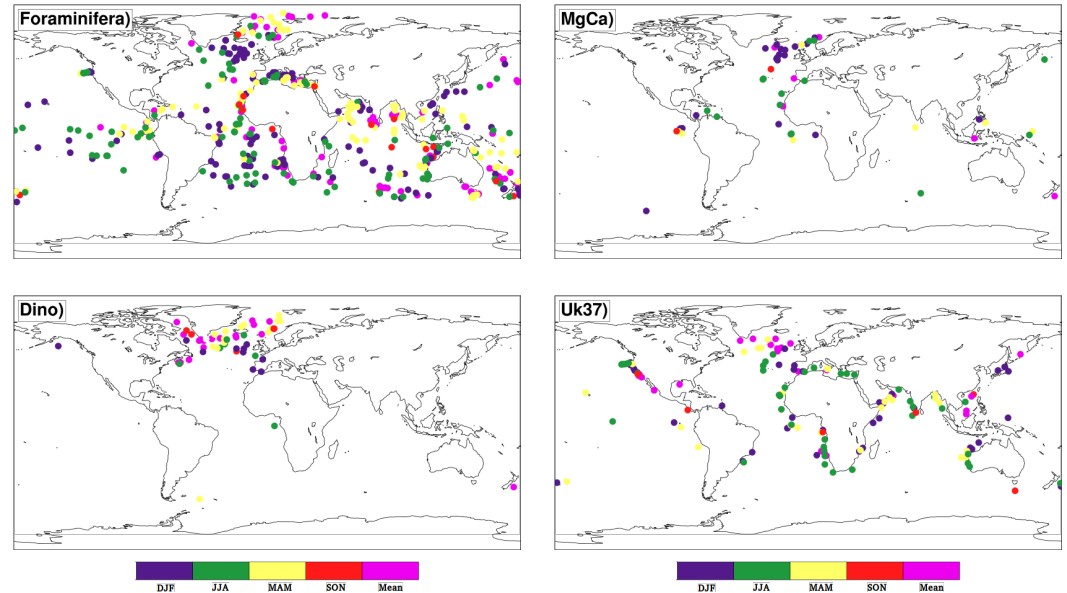

**Figure 6:** The colors fill of the circles represent the seasonal/annual mean in which the reconstruction of foraminifera, MgCa, dinoflagellates and $U^k_{37}$ records agree best with ensemble median of all PMIP3 models.

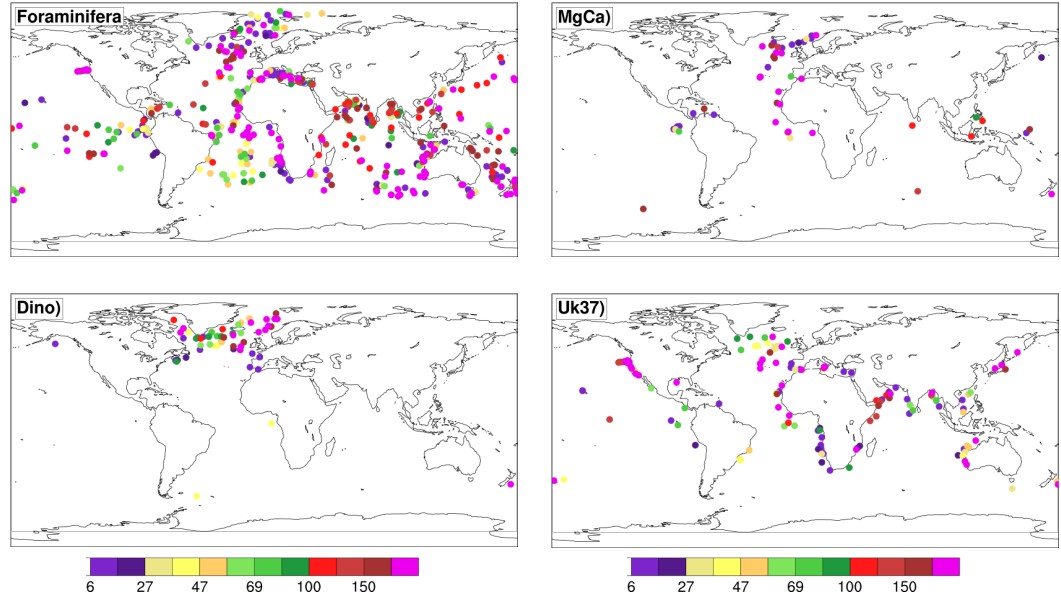

**Figure 7:** The colors fill of the circles represent the different depth (m) mean in which the reconstruction of foraminifera, MgCa, dinoflagellates and $U^k_{37}$ records agree best with ensemble median of all PMIP3 models.

770                                            **Tables**



**Table 1:** Correlation and RMSE between COSMOS LIS models annual mean SST and MARGO project dataset and MARGO proxies annual mean SST

|  | MARGO | Foraminifera | MgCa | Dinoflagellates | $U^{k}_{37}$ |
|---|---|---|---|---|---|
|  | R, RMSE | | | | |
| LGMctl | 0.14, 3.16 | 0.43, 2.68 | 0.51, 5.93 | -0.23, 6.43 | -0.04, 3.42 |
| Gowan | 0.10, 3.10 | 0.44, 2.54 | 0.64, 5.92 | -0.26, 6.64 | -0.09, 3.48 |
| Ice6g | 0.15, 3.54 | 0.47, 2.59 | 0.61, 5.91 | -0.24, 6.61 | -0.15, 3.54 |
| Lambeck | 0.15, 3.08 | 0.44, 2.61 | 0.57, 5.93 | -0.17, 6.36 | -0.05, 3.36 |
| Licc | 0.10, 3.12 | 0.45, 2.53 | 0.72, 5.89 | -0.22, 6.72 | -0.15, 3.68 |
| Tarasov | 0.16, 3.15 | 0.48, | 0.68, | -0.22, | -0.17, |
| Median_LIS | 0.15, 3.08 | 0.48, 2.58 | 0.62, 5.90 | -0.23, 6.69 | -0.12, 3.57 |

**Table 2:** Correlation and RMSE between best-fit season of COSMOS LIS SST and proxies annual mean SST

|  | Foraminifera | MgCa | Dinoflagellates | $U^{k}_{37}$ |
|---|---|---|---|---|
|  | R, RMSE | | | |
| LGMctl | 0.54, 2.52 | 0.59, 5.85 | 0.01, 4.88 | 0.19, 3.28 |
| Gowan | 0.59, 2.18 | 0.73, 5.68 | -0.01, 5.74 | 0.19, 3.18 |
| Ice6g | 0.59, 2.24 | 0.67, 5.67 | 0.00, 5.72 | 0.14, 3.26 |
| Lambeck | 0.57, 2.15 | 0.63, 5.66 | 0.09, 5.81 | 0.21, 3.27 |
| Licc | 0.60, 2.26 | 0.75, 5.70 | 0.01, 5.50 | 0.15, 3.12 |
| Tarasov | 0.60, 2.22 | 0.67, 5.67 | 0.01, 5.78 | 0.13, 3.29 |
| Median_LIS | 0.59, 2.22 | 0.67, 5.67 | 0.02, 5.74 | 0.16, 3.25 |


**Table 3:** Correlation and RMSE between best-fit depth of COSMOS LIS SST and proxies annual mean SST

|  | Foraminifera | MgCa | Dinoflagellates | $U^{k}_{37}$ |
|---|---|---|---|---|
|  | R, RMSE | | | |
| LGMctl | 0.62, 2.10 | 0.65, 5.72 | 0.16, 4.57 | 0.36, 2.90 |
| Gowan | 0.65, 2.06 | 0.75, 5.71 | 0.39, 4.15 | 0.38, 2.85 |
| Ice6g | 0.64, 2.06 | 0.72, 5.69 | 0.11, 4.70 | 0.27, 3.03 |
| Lambeck | 0.65, 2.05 | 0.72, 5.71 | 0.08, 4.74 | 0.31, 2.95 |
| Licc | 0.66, 2.04 | 0.78, , 5.69 | 0.37, 4.24 | 0.33, 2.92 |
| Tarasov | 0.65, 2.05 | 0.71, 5.84 | 0.08, 4.78 | 0.28, 3.02 |
| Median_LIS | 0.65, 2.05 | 0.72, 5.70 | 0.15, 4.63 | 0.31, 2.95 |