# Peer review of "A model-data comparison of the Last Glacial Maximum surface temperature changes"

_Climate of the Past, 2018_

## Author Comment (AC1) · 7 Mar 2018

Dear Editor and Reviewers,

The corrected version of all figures and tables are uploaded here in supplements. Please check it out.

Regards Akil Hossain

Please also note the supplement to this comment:
https://www.clim-past-discuss.net/cp-2018-9/cp-2018-9-AC1-supplement.pdf

[Figure]

[Figure]

**Figure 2:** Global reconstructed SST trends of four proxies compared to simulated annual mean SST anomalies as calculated from the ensemble median mean of the models listed in PMIP3 (a) and IPSL-CM5A-LR (b). The colors fill of the circles show the temperature anomalies as recorded by MARGO proxy records. Background color fill: simulated global pattern of annual mean sea surface temperature changes between the LGM and PI climate.

**Fig. 1.** Figure 2

[Figure]

**Figure 3:** Global SST anomalies of the annual mean LGM LIS experiment output, and temperature trends based on planktonic foraminifera, Mg/Ca, dinocyst, alkenones ($U^k_{37}$), diatoms and radiolarian reconstructions of MARGO project. Background color fill: simulated annual mean SST changes between the Tarasov_LIS and PI. The circles localize the three different proxy records, respectively.

**Fig. 2.** Figure 3

[Figure]

Figure 5: (a) The circles localize the foraminifera, MgCa, dinoflagellates and U$^k_{37}$ records and the colors fill of the circles represent the different depth (m) mean in which the reconstruction agrees best with model. (b) Background color fill: simulated global pattern of annual mean sea surface temperature changes between the LGM and PI climate. Colors fill of the circles show the temperature change recorded by corresponding depth mean shown in (a) at the sample locations.

**Fig. 3.** Figure 5

---

## Referee Comment (RC1) · Anonymous Referee #1 · 6 Apr 2018

Review of Hosain et al. 'A model-data comparison of the Last Glacial Maximum surface temperature changes'. The authors show a comparison between reconstructions of LGM temperature on land and in the ocean and several climate model simulations. They use two different compilations of temperature data and pay special attention to potential recording biases in the marine proxy data. The authors also use several configurations of the COSMOS model with different ice sheets and the PMIP3 models. Hosain et al show that particularly in the marine realm there is considerable mismatch between the data and the models and they suggest that these are due to seasonal and depth recording biases in the proxies.

The paper has an ambitious two-fold goal: i) to assess the different ice sheet reconstructions and ii) to assess biases in the recording of (marine) proxies. Both are impor-

tant questions. However, after reading the manuscript it is still unclear to me why and how the ice sheet reconstruction by Tarasov is better and what we have learned about marine temperature proxies apart from the known fact that they might be biased to variable seasons and depths. A paper that explicitly states 'comparison [of proxies] with outputs from climate model will help to understand the recording system itself' (L73) should deliver more and provide new insights, or directions, into how we can overcome the known recording biases. The approach taken by the authors is to simply look at what depth or season the marine proxy system correlates best. This implies that the recording bias may vary randomly from site to site. While there is nothing wrong with that approach as a starting point, we know that the ecology of the proxy carriers is not random (see e.g. the discussion section on alkenones or Leduc et al. [2010] or Jonkers and Kucera [2015]). The offsets between the annual mean SST and the reconstructed SST are thus likely to follow a systematic trend, likely with temperature. Rather than showing that ecology leaves an imprint on proxies (which is old news) the authors should investigate whether they see such trends in their comparison. A model that shows a pattern in the offset that is consistent with our understanding of the ecology of the recorder could arguably be considered to have more skill than one that doesn't. The opposite (no pattern, or random deviations) are more likely to be related to simple noise in the reconstructions or models. In this way models and data can be more meaningfully compared and new insights about the recording systems might be obtained. Related to this, it remains unclear how depth and season in the recording bias are separated? The same temperature can often be found at different times of the year or at different depths. How is this dealt with in paragraph 4.4? And what season is assumed in paragraph 4.3? In addition, why is seasonal recording not considered for the terrestrial proxies? And is it right that the evaluation of the different ice sheet topographies in based solely on the terrestrial data? I couldn't find a figure or table with summary statistics.

Equally importantly, the comparison between the reconstructions and the models could be improved. A simple correlation can be very misleading and the RMSE (deviation

from the 1:1 line, why in per mille?) is a much more useful measure of the difference. Moreover, there is no statistical treatment of the uncertainties in the data or the model (at the minimum interannual variability in the model and the reported errors on the reconstructions should be taken into account). None of the statements about significance are accompanied by an explanation how this was determined and at what confidence level. This leaves the reader wondering whether the differences between the different ice sheet configurations or the different season/depth biases are real or meaningful. This is crucial as many differences between the models are very small.

At some places in the manuscript the authors mention uncertainty in the models too. It would be good if they discuss this more upfront. With so many models and different configurations of the same model (in this case the ice sheet topography) there are many degrees of freedom and there is a large chance of being right for the wrong reasons, not only because the proxies are biased (L163). How do the authors deal with that? Related to this, what have we learned about the model (configuration)? If some of the observed differences between the model runs are real/significant, then why? Where? Can the authors go deeper into the mechanisms or the physics that explain the differences?

In addition, the manuscript lacks a clear separation between results and discussion and the discussion section itself does hardly discuss the results, but rather summarises what others have said about potential recording biases in marine proxies. A lot of this could be placed in the introduction instead. Finally, there are numerous spelling and style errors. I have indicated some in the line-by-line comments below, but I recommend that the authors thoroughly proofread a revised version. I am sorry that I am unable to provide better news at this time, but I hope that my comments will help to improve the manuscript.

Line by line comments

L8: 'abrupt'. Reconsider wording What is meant here?

[Figure]

L11-12: reword ' . . .pollen and plant macrofossil based. . .'

L16: it is the simulation using the Tarasov reconstruction that shows the highest correlation, not the reconstruction.

L33: Project instead of Projection

L40: please be more specific, uncertainty of what?

L54: please add a sentence or two to explain the link between the beginning and end of this paragraph. Importantly, Jonkers and Kucera [Jonkers and Kučera, 2017] –and before them several others [e.g. Mix, 1987; Schmidt, 1999; Schmidt and Mulitza, 2002; Skinner and Elderfield, 2005] – showed that there is predictability in the recording bias. This is an important point as it may help to distinguish between different models and or estimates of recording depth/season.

L74: replace 'will help' with 'might help'

L76: 'can force' – consider rewriting. Also, rewrite statement about all models in the next sentence. The PMIP3 ensemble does not contain all models of LGM climate.

L78: Strictly speaking there is no ecological effect on the proxy interpretation, there is an ecological effect on the recording of the climate sensor (proxy) [see for instance Evans et al., 2013].

L95: is Zhang et al. 2013 appropriate for the PMIP3 protocol?

L109-134: what exactly is compared, the gridded products of the reconstructions or the individual sites? If the latter, why is the gridding explained and how were the data compared precisely?

L148: positions of brackets is incorrect.

L166-174: this is discussion. No references in results section.

L195: Change to 'Proxy-specific comparison' or equivalent.

L211: add uncertainties in the transfer functions. Or non-temperature effects on the assemblages?

L231-240: discussion. It is also unclear to me what the main message of this paragraph is.

L252: R = 0.01 means no correlation, not a positive one.

L256: the data is not composed of planktonic organisms, it's based on measurements of their fossil remains. Also reword 'shift in the different water columns'.

L260: Coccolithophores (the alkenone-producing organisms) are phytoplankton and require light for photosynthesis. The same holds for other phytoplankton and symbiont-bearing planktonic foraminifera. 183 m seems rather deep for phytoplankton. I assume that light availability is not modelled, but the authors should look into this and assess whether the inferred recording depths (e.g. L269) are consistent with the ecology of the proxy carriers. There is also a lot of discussion in these sections.

L270-274: this sentence begins and ends with different statements about the habitat depth of planktonic foraminifera. Please explain the difference, or discuss it. See also Rebotim et al. [2017] for a discussion on the variability of depth habitat.

L289-295: I disagree, if the data and the model disagree, and consistently disagree the reason is unlikely to be due to uncertainty in the data alone. Uncertainty in the data would lead to random variations around the mean value, not indicate consistent (temporal/spatial) changes. It is more likely that the mismatch is due to uncertainties/unknowns in both the data and the models. It would be good if the authors acknowledge that more.

L327-329: this section on sediment traps needs referencing. It is also well known that there is no uniform seasonality of planktonic foraminifera, rather seasonality varies spatially [Jonkers and Kučera, 2015; Tolderlund and Bé, 1971] and has hence likely varied in the past.

L336-337: please be specific: uncertainty for planktonic foraminifera proxies, not the foraminifera themselves. Moreover, this not only holds for planktonic foraminifera, but for all proxy carriers with a short (< 1 year) life span [e.g. for coccolithophores that produce the alkenones Rosell-Melé and Prahl, 2013].

L344-357: so it seems that there is a pattern in the season that is preferably reflected in the UK37 ratio. Is this resolved in the model-data mismatch? Does any model yield data more consistent with such a pattern? It is this kind of analysis that is lacking from the present manuscript.

L364: proxies are not exposed to nutrient conditions, the organisms are.

L377: Deuser and Ross and Anand et al used the same sediment trap time series for their analysis, so this is only regionally constrained information. Crucially, one cannot infer living depth from sediment traps (perhaps the authors mean calcification depth).

L380-384: this idea is hardly new, Emiliani [Emiliani, 1954; 1955] already touched on this. Please include.

L391: it is unclear what is meant with 'in such a way'.

L395: There is also observational data that shows the dampening effect of changing habitat of the proxy carrier [Ganssen and Kroon, 2000; Jonkers and Kučera, 2017].

L400: why on the contrary, I don't understand the difference. And please explain why it is important to model foraminifera growth, rather than abundance. Note also that Fraile et al used many more variables than temperature alone [Fraile et al., 2008] (in fact, more than Lombard) and see Kretschmer et al [Kretschmer et al., 2017] for an update of this model.

L406-412: I think a more upfront discussion of inherent uncertainties in the model is essential and should be placed not at the end of the discussion and include more than just model resolution.

L420-421: Sentence incomplete or wrong.

L423-427: this fundamental mismatch between the models and the data is mentioned here for the first time. It deserves mentioning in the results and discussion. As to the question whether it is the models or the data that cause this discrepancy, it is important to note that our current understanding of proxy carriers (in particular planktonic foraminifera) is that they tend to underestimate the environmental change (see suggested references and studies cited in the manuscript). Such homeostatic behaviour only exacerbates the mismatch.

Fig. S1 is directly copied from the MARGO paper, I don't know if this is appropriate with regards to copy rights etc.

Table 1: why is there no RMSE for the Tarasov reconstruction? Also, none of the errors have units. Similarly, the legends in the figures often lack units.

References: Emiliani, C. (1954), Depth habitats of some species of pelagic Foraminifera as indicated by oxygen isotope ratios, Am J Sci, 252(3), 149-158.

Emiliani, C. (1955), Pleistocene temperatures, The Journal of Geology, 538-578.

Evans, M. N., S. E. Tolwinski-Ward, D. M. Thompson, and K. J. Anchukaitis (2013), Applications of proxy system modeling in high resolution paleoclimatology, Quaternary Science Reviews, 76(0), 16-28.

Fraile, I., M. Schulz, S. Mulitza, and M. Kucera (2008), Predicting the global distribution of planktonic foraminifera using a dynamic ecosystem model, Biogeosciences, 5(3), 891-911.

Ganssen, G. M., and D. Kroon (2000), The isotopic signature of planktonic foraminifera from NE Atlantic surface sediments: implications for the reconstruction of past oceanic conditions, Journal of the Geological Society, 157(3), 693-699.

Jonkers, L., and M. Kučera (2015), Global analysis of seasonality in the shell flux of

extant planktonic Foraminifera, Biogeosciences, 12(7), 2207-2226.

Jonkers, L., and M. Kučera (2017), Quantifying the effect of seasonal and vertical habitat tracking on planktonic foraminifera proxies, Clim. Past, 13(6), 573-586.

Kretschmer, K., L. Jonkers, M. Kucera, and M. Schulz (2017), Modeling seasonal and vertical habitats of planktonic foraminifera on a global scale, Biogeosciences Discuss., 2017, 1-37.

Leduc, G., R. Schneider, J. H. Kim, and G. Lohmann (2010), Holocene and Eemian sea surface temperature trends as revealed by alkenone and Mg/Ca paleothermometry, Quaternary Science Reviews, 29(7-8), 989-1004.

Mix, A. (1987), The oxygen-isotope record of glaciation, The Geology of North America, 3, 111-135.

Rebotim, A., A. H. L. Voelker, L. Jonkers, J. J. Waniek, H. Meggers, R. Schiebel, I. Fraile, M. Schulz, and M. Kucera (2017), Factors controlling the depth habitat of planktonic foraminifera in the subtropical eastern North Atlantic, Biogeosciences, 14(4), 827-859.

Rosell-Melé, A., and F. G. Prahl (2013), Seasonality of temperature estimates as inferred from sediment trap data, Quaternary Science Reviews, 72(0), 128-136.

Schmidt, G. A. (1999), Forward modeling of carbonate proxy data from planktonic foraminifera using oxygen isotope tracers in a global ocean mode, Paleoceanography, 14, 482-497, doi:410.1029/1999PA900025.

Schmidt, G. A., and S. Mulitza (2002), Global calibration of ecological models for planktic foraminifera from coretop carbonate oxygen-18, Marine Micropaleontology, 44(3-4), 125-140.

Skinner, L. C., and H. Elderfield (2005), Constraining ecological and biological bias in planktonic foraminiferal Mg/Ca and $\delta$18Occ: A multispecies approach to proxy calibration testing, Paleoceanography, 20(1), n/a-n/a.

Tolderlund, D. S., and A. W. H. Bé (1971), Seasonal distribution of planktonic foraminifera in the western North Atlantic, Micropaleontology, 17(3), 297-329.

---

## Referee Comment (RC2) · Anonymous Referee #2 · 4 May 2018

I am grateful for the opportunity to review the article entitled "A model-data comparison of the Last Glacial Maximum surface temperature changes" by A. Hossain and co-authors. This study simulates the temperature changes during the Last Glacial Maximum with the GCM model COSMOS and compare the outputs with the temperature reconstructed from both marine and terrestrial data (Bartlein et al., 2011 and MARGO datasets). They use six different Laurentide Ice Sheet reconstructions in the COSMOS model. Simulations based on PMIP3 and IPSL models are also compared with the proxies-inferred SSTs. They evidenced some mismatches between simulated and reconstructed SSTs and investigate the role of the seasonality and the depth as potential bias in the proxies to explain such discrepancies.

I think that the paper of A. Hossain et al. presents interesting findings in terms of results

to be published in C. Past but I also think that it cannot be published in its current version. My first point concerns the structure of the paper which is not appropriate as it is. In draft I have, a lot of things are confusing (for ex., figures 6 et 7 are not mentioned and discussed in the text; it's quite the same for the figure with PMIP and IPSL models even if I have downloaded the corrected version). I have the feeling that the paper has been written fast and a considerable effort of rewriting is needed to improve the quality of the draft before to be accepted. The structure of the results and discussion parts is not clear and not easy to follow. I strongly recommend to group the results and the discussion in the same part. I propose to restructure it as follow: 1. model comparison 1.1 LIS simulations 1.2 COSMOS, PMIP3, IPSL comparison 2 data-model comparison: 2.1 terrestrial temperatures changes 2.2 SSTs changes (including seasonality and depth) The comparison with the terrestrial temperature changes is an important point given that if I remember well, PMIP models often underestimated the LGM cooling. SO I would like to add this figure not in the supplementary material as it is in the current form but in the text to support the discussion. May be you can also compare the seasonal parameters (temperature of the coldest and warmest month) simulated by COSMOS with MTCO and MTWA inferred from pollen.

My second point concerns the originality of this paper. The objective and the questions of your paper need to be better justified. There is a lot of simulations on the LGM as this period has been chosen by the PMIP modelers; therefore we need to better understand what are your questions, and what is new compared to previous studies. I have the same feeling for data-model comparisons: they are a lot of, and the Bartlein and Margo datasets are not new: could you better clarify the originality of your approach and of your results?

Moreover, several points of the discussion must be discussed more in depth or are lacking (see below).

Other points:

-Title: should be more informative - Abstract : must be more precise (key results, conclusions) - Introduction : -line 32 : correct "Projection" with Project"; more refs are needed for PMIP (Joussaume et al. . .); -Line 36: You state that "Previous studies proposed that these northern hemisphere ice sheets, especially the North American Laurentide ice sheet (LIS), are of crucial importance on modulating glacial climate": could you briefly precise its role on glacial climate? -Line 51 and 110 : Please avoid to use the term "plant macrofossils": the annual temperature is mainly based on pollen data through transfer function (sites with macrofossils data are very few for the LGM and its complex to provide robust quantitative temperature estimates from macrofossils alone). -Line 63: I don't think the ref Davis and Brewer (2009) is appropriate here, they don't talk about the bloom season; -Line 76: a ref is needed for the model COSMOS

- Material and methods - a table with the different LIS reconstructions will be welcome -line 111: you mention the different climate parameters reconstructed from pollen data in the Bartlein et al synthesis (MTWA, GDD5 MTCO and MAT). As you only use in your paper the annual temperature reconstruction, it's not necessary to mention the others climate parameters. In contrast, you can write few words on the method used (Mat, transfer function or inverse modelling). I strongly recommend to also compare the seasonal parameters (temperature of the coldest and warmest month) simulated by COSMOS with MTCO and MTWA inferred from pollen. -line 136: avoid words as Âń proxy-derived observational data Âż; the temperature is reconstructed from proxies, it's not observational data. Use instead proxy-inferred temperature; there is a lot of such approximations in the text, please correct it everywhere.

- Results: not appropriate as it is. I also strongly recommend to group the results and the discussion in the same part to avoid to be lost. -line 225: you state that In the North Atlantic Ocean, the best agreement of planktonic foraminifera, dinoflagellates, and alkenones is found for local summer. I don't agree with you, I don't see it on the figure. -line 259: how did you define the different layers? Arbitrary or statistical threshold? -line 264: why 37m? not 27 or 47as in the caption? -line 305: what do

you mean by "Instead, the ensemble median (Fig. 2a) typically displays the common signal. In this case, it is the mean value of the fourth and fifth ensemble member out of eight models which are ordered according to ranked values"? Really not clear!

- Discussion is too short, it must be clarified according to the objectives of the paper and the results (which also need to be more precise). A comparison between these results and previous LGM simulations is required and must be discussed. A comparison with the CLIMAP values and other studies will also be welcome. A more in depth discussion of the agreement between terrestrial pollen-based temperature and model output is also needed as I have in mind that usually the models underestimates the values inferred from the data as it was the case in the previous PIMP simulations. -in the 4.3 part, you discuss foraminifers, alkenones and MGCa ratio, but nothing is written about the dinos. I'm sure that a lot of papers are available. - Data model discrepancies can also been explained by the proxy itself or by the method (transfer function...); this point is important and need to be discussed - Line 410: You state that the proxy records used in most of the studies are more often located in coastal areas. I don't agree with you: Dino and forams records are not only located in coastal areas.

- Figures : the order of each figure must be carefully checked in the text. The colors of the figures 5 a and 7a must be changed for more clarity

I think that there is potential in this paper in terms of results; however a considerable effort of rewriting is needed to improve the manuscript before acceptation.

Please also note the supplement to this comment:
https://www.clim-past-discuss.net/cp-2018-9/cp-2018-9-RC2-supplement.pdf

---

## Author Comment (AC2) · 9 May 2018

Against the Review comments: 'Fig. S1 is directly copied from the MARGO paper, I don't know if this is appropriate with regards to copy rights etc.'

Authors comment: We have the permission from Nature Geoscience to reuse the Fig. S1. This figure has been uploaded with revised caption.

[Figure]

**Fig. 1.** Fig. S1: Distribution of MARGO data points, indicating also which proxy was measured at each location (Waelbroeck et al., 2009 ©Nature Geoscience).

Legend:
- ☆ Diatoms
- △ Dinoflagellates
- ○ Foraminifera
- ▽ Radiolaria
- ▢ Mg/Ca
- + $U_{37}^{K'}$

---

## Author Comment (AC3) · 7 Aug 2018

Answer to reviewers' comments: A model-data comparison of the Last Glacial Maximum surface temperature changes

Akil Hossain, Xu Zhang, Gerrit Lohmann

Alfred Wegener Institute, Helmholtz Centre for Polar and Marine Research, Bremerhaven, Germany.

General remarks

We are very thankful to the editor and reviewers for the effort and time dedicated to the reviewing of our manuscript and for the helpful reviews. In order to address all

the concerns raised by the reviewers we have significantly restructured the manuscript and few new section. In this document, we supply detailed responses to all comments, suggestions and notes made by the two reviewers. We hope that the applied revisions are to the satisfaction of the reviewers and the editor.

Reviewer #2: Comment R2.1

My first point concerns the structure of the paper which is not appropriate as it is. In draft I have, a lot of things are confusing (for ex., figures 6 et 7 are not mentioned and discussed in the text; it's quite the same for the figure with PMIP and IPSL models even if I have downloaded the corrected version).

AC(Author Comment): Structure of the paper is changed and discuss in detailed at the Comment 2.2. These issue with figure has been revised and solved.

Author's changes in manuscript: Figure 6 is now mentioned in the section 3.2.1 and Figure 7 is removed according to our new structure. Figure with PMIP and IPSL models also now mentioned in the section 3.2.1.

Comment R2.2

The structure of the results and discussion parts is not clear and not easy to follow. I strongly recommend to group the results and the discussion in the same part. I propose to restructure it as follow: 1. model comparison 1.1 LIS simulations 1.2 COS-MOS, PMIP3, IPSL comparison 2 data-model comparison: 2.1 terrestrial temperatures changes 2.2 SSTs changes (including seasonality and depth)

AC: By considering this suggestion our result and discussion part is restructured as follows:

3.1 Data Model Comparison: COSMOS LIS reconstructions 3.1.1 Sea surface temperature changes 3.1.2 Proxy-specific comparison of SST Annual Mean 3.1.3 Seasonality of the recorder system 3.1.4 Land Surface temperature changes 3.1.5 Mean temperature of coldest and warmest month 3.2 Data Model Comparison: PMIP3 models 3.2.1

Sea surface temperature changes 3.2.2 Land Surface temperature changes

Considering habitat depth of the planktonic organisms make our manuscript more complicated and there are many debates about habitat depth of the organisms, therefore, according to our new structure, we have removed the habitat depth analysis of proxies. So this section is no more in the manuscript.

Author's changes in manuscript:

Removed–

L21: habitat depths and L23: habitat depth and L55: as well as habitat depth L62-65: Thereby, two parameters might influence the estimations of LGM SST anomalies: changes in bloom seasons (Rosell-Melé et al., 1995; Sikes et al., 1997; Barker et al., 2005; Haug et al., 2005; Davis and Brewer, 2009; Fraile et al., 2009; Kim et al., 2015), and changes in habitat depth of different species (Nürnberg et al., 1995; Bentaleb et al., 1999; Barker et al., 2005; de Vernal et al., 2006; Kim et al., 2015).

L66: water depths and L70: habitat depth and L72: and water depths L163: and depth L255-88: 3.5 Habitat depth of the recorder system

In this study, the observational data of MARGO project is composed of different Planktonic organisms which are however known to be able to move in the different water column (e.g. Conte et al., 2006). In order to observe whether deeper layers in the model would be in better agreement with the temperature reconstruction than surface, the model for different layers of the upper 183 m of the ocean were compared to the proxy records (Fig. 5a). Layers below these depths can be ignored, since alkenone-producing organisms require sun-light for photosynthesis.

From the Fig. 3, it can be observed that most of the core sites in the North Atlantic are in best agreement with the surface layers (0 to 37 m) and in the Southern Hemisphere with subsurface layers (between 70 to 183 m). For dinoflagellates, most core point of it also agree with subsurface layer. For the Mg/Ca ratio, the same upper layers are

also considered, but from the Fig. 5a it can be seen that a big part of the record best agree with the upper level of the ocean although a few cores in the North Atlantic from subsurface layers. For alkenones, a big part of the records best agree with the deeper level of the ocean, between 70 and 183 m (Fig. 5a). Rest of the cores do not show any clear evidence for a preferred ocean layer (Fig. 5a).

Investigations of foraminiferal Mg/Ca distributions in the Atlantic, Pacific, and Indian Oceans have found that 75–100% of foraminifera species live between 0 to 40 m water depths (van Donk 1977; Kim et al., 2015). Alkenone-based temperatures record do not always precisely represent the SST, because of ambiguities due to the habitat depth of alkenone-producing organisoms (Ohkouchi et al. 1999; Lee and Schneider 2005). The planktonic foraminifera live over a range of depths, mostly between about 75 and 250 m water depth (de Vernal et al., 2006).

The correlation between the reconstructed and simulated anomalies has been also increased significantly for best fit depth. A higher correlation has been found for the best fit depth (R = 0.65) than the annual mean (R = 0.51) in case of Mg/Ca (Table 1 and Table 3). Foraminifera show also significantly high correlation for depth (R = 0.54) which is also higher than annual (R = 0.43). A positive correlation have been found for the best fit depth in case of dinoflagellates and alkenones (dinoflagellates, R = 0.16; alkenones, R = 0.36) (Table 3) where they show negative correlation for annual mean (Table 1).

If we consider best fit seasons and depths together at the same time at a specific data point in the model simulations, the mismatch between the climate simulation of the model and different proxy records has been decreased and correlation coefficient has been also increased significantly (Fig.: S4; Table S5). Planktonic foraminifera inhabit a wide range of the water column and also show variability in their seasonal abundance. The average proxy signal in sedimentary foraminifera is weighted towards conditions at the season and depth of calcification rather than reflecting annual mean surface conditions (Jonkers and Kučera, 2017).

L309: habitat depth,. . .. . ., and habitat depth-seasonality together (new L275) L319: and depth (new L285) L324: water depth and (new L290) L360-78: 4.3 Habitat depth biases

Changes in the habitat depth of the SST over the LGM result in deviations between model simulations and proxy records. Such changes in recording season and habitat depth could have been caused by changes in insolation over the LGM or by related changes in the nutrient distribution and ocean temperature that the individual proxies are exposed to (Lohmann et al., 2013). Comparing the reconstructed LGM temperature trends at model levels in the upper 183 m does not remove the discrepancy between models and proxies. In comparison with the previous study indicates the planktonic foraminifera live over a range of depths, mostly between about 75 and 250 m water depth (de Vernal et al., 2006). A recent study (Kim et al., 2015) suggested that foraminiferal Mg/Ca record temperatures at depths of 0-40 m in the water column. Alkenone concentrations were high at the subsurface water column of the central Pacific Ocean (Lee and Schneider, 2005) agree with model simulation. The majority species of the planktonic foraminifer assemblages in polar waters inhabits subsurface waters (Bé and Tolderlund, 1971) and may find below the thermocline as it is frequently the case in the Arctic, subarctic and other stratified waters where it occurs along or below the halocline (Kohfeld et al., 1996; de Vernal et al., 2002; Hillaire-Marcel et al., 2004; de Vernal et al., 2006). Planktonic foraminifera normally migrate vertically throughout their life cycle, forming calcite at deeper layers as they mature. This temperature/depth migration results in heterogeneity of Mg/Ca ratios within tests of individual foraminifera (Nürnberg, 1995; Jha and Elderfield, 2000; Elderfield and Ganssen, 2000; Benway et al., 2003). It has been observed that various planktonic species add a calcite crust in colder, deeper water immediately prior to reproduction (Barker et al., 2005). Many species of planktonic foraminifera live at depths greater than 50m (Erez and Honjo, 1981; Deuser and Ross, 1989; Anand et al., 2003) which agree with the studied model simulation.

L424: and different water depths at which the recording organisms may have lived
L429-32: Comparing the reconstructed LGM temperature anomalies with the model levels of the upper 183 m does not remove the discrepancy between models and proxies. For Mg/Ca ratios, a large number of records fit best with the model surface layer. In case of the remaining three proxies, the best-fit agreement is found in the subsurface layer. Thus, we find mostly the highest agreement between proxy-record and simulated temperature anomalies in the subsurface layer depth.

Added–

L62-65: Thereby, changes in bloom seasons might influence the estimations of LGM SST anomalies (Rosell-Melé et al., 1995; Sikes et al., 1997; Barker et al., 2005; Haug et al., 2005; Davis and Brewer, 2009; Fraile et al., 2009; Kim et al., 2015).

L243: 3.1.5 Mean temperature of coldest and warmest month

According to Bartlein et al. (2011), July temperature in the northern hemisphere (southern hemisphere - December) has been combined with reconstructions of mean temperature of the warmest month (MTWA). Similarly, December temperature in the northern hemisphere (southern hemisphere - July) has been combined with reconstructions of mean temperature of the coldest month (MTCO) (Fig. S3, see also Bartlein et al., 2011).

During the LGM, Africa show warmer (1 to 4°C) than today in the reconstruction of MTWA (Fig. S3, see also Wu et al. 2007). A few sites in the northern hemisphere especially in the Alaska, reconstruction of warmer conditions as shown by seasonal temperature variable MTWA and similar or slightly warmer than today is registered chiefly in MTCO (Fig. S3) (Bartlein et al., 2011). The LIS was large enough to cause atmospheric circulation pattern reorganization. This reorganization could have originated in more southerly landward flow into Alaska, that would have produced advective warming in this region year-round (Bartlein et al., 2011). In general, the summer temperatures changes as represented by MTWA (Fig. S3) are smaller than the winnone

ter temperatures changes as represented by MTCO (Fig. S3, see also Bartlein et al., 2011).

For a comparison with proxy data, the warmest and coldest months of the model results have been compared with the seasonal temperature variables MTWA and MTCO. For MTWA, the highest correlation coefficient and lowest deviations are found for the LGMctl (R = 0.50, RMSE = 6.5‰ and Ice6g_LIS (R = 0.50, RMSE = 6.5‰ ice-sheet reconstruction and the lowest correlation coefficient and largest deviations for the Gowan_NAIS (R = 0.44, RMSE = 6.3‰ (Fig. 5). Similarly, for MTCO, the highest correlation coefficient and lowest deviations are also found for the LGMctl (R = 0.46) and Ice6g_LIS (R = 0.46) and the lowest correlation coefficient for the Gowan_NAIS (R = 0.43) (Table 3). Overall, the correlation coefficient value for warmest and coldest months of the model has been increased than the model Annual mean value (Table 3).

L291: 3.2.2 Land Surface temperature changes

The annual mean SAT of PMIP3 LGM climate is on average 4.5 oC colder than the PI climate and CNRM is comparatively warmer (annual mean temperature -2.6 oC) than other models. PMIP3 model results have been compared with the LGM continental temperature reconstruction by Bartlein et al. (2011). The reconstructions show year-round cooling during the LGM over the continents except a few sites in Alaska (Fig. 7) (Bartlein et al., 2011). Similar as SST reconstructions, among the eight PMIP3 model, IPSL-CM5A-LR (R = 0.27, RMSE = 3.3‰ shows the highest correlation (Table S5), although most of the model show low correlation coefficient with the reconstructed data-set. MTWA (highest R is 0.53) show higher correlation than MAT and MTCO (highest R is 0.27 and 0.48). Overall, the correlation between model and data has been increased for MTWA and MTCO than the model Annual mean value (Table S5).

Comment R2.3

The comparison with the terrestrial temperature changes is an important point given that if I remember well, PMIP models often underestimated the LGM cooling. SO I

would like to add this figure not in the supplementary material as it is in the current form but in the text to support the discussion. May be you can also compare the seasonal parameters (temperature of the coldest and warmest month) simulated by COSMOS with MTCO and MTWA inferred from pollen.

AC: This figure of the comparison with the terrestrial temperature changes is shifted to the main script. A subsection of seasonal parameters comparison also added to the manuscript.

Author's changes in manuscript: Fig. S2 now Fig.5 See Comments 2.2 for the Paragraph "3.1.5 Mean temperature of coldest and warmest month"

Comment R2.4

My second point concerns the originality of this paper. The objective and the questions of your paper need to be better justified. There is a lot of simulations on the LGM as this period has been chosen by the PMIP modelers; therefore we need to better understand what are your questions, and what is new compared to previous studies. I have the same feeling for data-model comparisons: they are a lot of, and the Bartlein and Margo datasets are not new: could you better clarify the originality of your approach and of your results?

AC: Our submitted manuscript is examined the uncertainties in SAT and SST of different ice sheets reconstructions and the PMIP3 models which is unique in its kind of research.

The main objectives of our paper: i) to assess the different LIS reconstructions and PMIP3 models and ii) to assess biases in the recording of different proxies. We have compared the reconstructions of LGM temperature on land and in the ocean with the different ice sheets reconstructions and the PMIP3 models. We have also assess the potential recording biases in the proxy data and found that particularly in the marine realm there is considerable mismatch between the data and the models and these are

due to seasonal recording biases in the proxies.

There are some more datasets for temperature reconstruction but MARGO and Bartlein dataset are much intense with large coverage area.

Author's changes in manuscript: No change

Comment R2.5

Moreover, several points of the discussion must be discussed more in depth or are lacking (see below). Other points:-Title: should be more informative

AC: After discussing with the co-authors, we have decided that the title is appropriate according to our content and not to change it.

Author's changes in manuscript: No change Comment R2.6

- Abstract : must be more precise (key results, conclusions)

AC: We have make a revision on abstract.

Author's changes in manuscript: Abstract Over the Last Glacial Maximum (LGM, $\sim$21ka BP), the presence of vast Northern Hemisphere ice-sheets caused abrupt changes in surface topography and background climatic state. While the ice-sheet extent is well known, several conflicting ice-sheet topography reconstructions suggest that there is uncertainty in this boundary condition. The terrestrial and sea surface temperature (SST) of the LGM as simulated with six different Laurentide Ice Sheet (LIS) reconstructions in a fully coupled Earth System Model (COSMOS) have been compared with the subfossil pollen and marine temperature proxies reconstruction. The terrestrial reconstruction shows a similar pattern and in good agreement with model data. The SST proxy dataset comprises a global compilation of planktonic foraminifera, diatoms, radiolarian, dinocyst, alkenones and planktonic foraminifera Mg/Ca-derived SST estimates. Significant mismatches between modeled and reconstructed SST have been observed. Among the six LIS reconstructions, simulation using Tarasov's LIS

reconstruction shows the highest correlation with reconstructed terrestrial and SST. Dinocyst-based SST records are much warmer than reconstructed by other proxies as well as Pre-industrial (PI) temperature. However, there are large discrepancies between model and temperature recorded by different proxies. In the most densely sampled Nordic Sea, the reconstructed LGM SST have large uncertainties. Eight different PMIP3 models also compared with temperature proxies reconstruction which show mismatches with the the proxy records might be due to misinterpreted and/or biased proxy records. Therefore, it has been speculated that considering different growing seasons of the planktonic organisms used for SST reconstruction could provide a better agreement with model results on a regional scale. Moreover, it can reduce model-data misfits. It is found that shifting in the living season can remove parts of the observed model-data mismatches in SST anomalies.

Comment R2.7

- Introduction : -line 32 : correct "Projection" with Project"; more refs are needed for PMIP (Joussaume et al...);

AC: This term has been corrected and reference has been added.

Author's changes in manuscript: Paleoclimate Modeling Intercomparison Project (PMIP). Joussaume and Taylor, 1995; Braconnot et al., 2007, 2012; Kageyama et al., 2006, 2013, Wang et al., 2013

Comment R2.8

-Line 36: You state that "Previous studies proposed that these northern hemisphere ice sheets, especially the North American Laurentide ice sheet (LIS), are of crucial importance on modulating glacial climate": could you briefly precise its role on glacial climate?

AC: The role of LIS on glacial climate is shortly added to the manuscript.

Author's changes in manuscript: L36: Changes in the ice sheet height can influence

on the ocean circulation and even the background climate (Zhang et al., 2014). The greater topography of the LISs forces a relatively stronger AMOC (Zhang et al., 2014).

Comment R2.9

-Line 51 and 110 : Please avoid to use the term "plant macrofossils": the annual temperature is mainly based on pollen data through transfer function (sites with macrofossils data are very few for the LGM and its complex to provide robust quantitative temperature estimates from macrofossils alone).

AC: This term has been removed from the manuscript.

Author's changes in manuscript: L51 and L110: "plant macrofossils" is romeved (new L52 and L109) Comment R2.10

-Line 63: I don't think the ref Davis and Brewer (2009) is appropriate here, they don't talk about the bloom season; AC: Yes, I partly agree with you as it doesn't mentioned directly bloom season but they used a palaeoclimate record to determine orbital forcing is biased towards the summer season and high latitudes.

Author's changes in manuscript: L64: 'Davis and Brewer, 2009' reference is removed. Comment R2.11

-Line 76: a ref is needed for the model COSMOS

AC: A reference Zhang et al., 2013 has been added. Author's changes in manuscript: (Zhang et al., 2013) Comment R2.12

- Material and methods - a table with the different LIS reconstructions will be welcome

AC: A table with the different LIS reconstructions and their references has been added.

Author's changes in manuscript: Table S1: LIS reconstructions used in this study.

Comment R2.13

-line 111: you mention the different climate parameters reconstructed from pollen data

Interactive
comment

in the Bartlein et al synthesis (MTWA, GDD5 MTCO and MAT). As you only use in your paper the annual temperature reconstruction, it's not necessary to mention the others climate parameters. In contrast, you can write few words on the method used (Mat, transfer function or inverse modelling). I strongly recommend to also compare the seasonal parameters (temperature of the coldest and warmest month) simulated by COSMOS with MTCO and MTWA inferred from pollen. AC: Considering this comment, the seasonal parameters (temperature of the coldest and warmest month) simulated by COSMOS and PMIP3 model have been compared with MTCO and MTWA inferred from pollen. A subsection of seasonal parameters comparison added to the manuscript.

Author's changes in manuscript: See Comment R2.2

Comment R2.14

-line 136: avoid words as Âń proxy-derived observational data Âż; the temperature is reconstructed from proxies, it's not observational data. Use instead proxy-inferred temperature; there is a lot of such approximations in the text, please correct it everywhere.

AC: This term has been checked and revised in the manuscript.

Author's changes in manuscript: L136, 159 and 432: proxy-inferred temperature data

Comment R2.15

- Results: not appropriate as it is. I also strongly recommend to group the results and the discussion in the same part to avoid to be lost. -line 225: you state that In the North Atlantic Ocean, the best agreement of planktonic foraminifera, dinoflagellates, and alkenones is found for local summer. I don't agree with you, I don't see it on the figure.

AC: There was a fault in the color bar of the Figure 4 (a), where JJA is shown as MAM and vice versa. However, in the North Atlantic Ocean, best agreement for organisms is not very clear but highest amount of data was found representing summer temperature.

Author's changes in manuscript: The color bar of the Figure 4 (a) is changed. Comment R2.16

-line 259: how did you define the different layers? Arbitrary or statistical threshold? -line 264: why 37m? not 27 or 47as in the caption?

AC: Considering habitat depth of the planktonic organisms make our manuscript more complicated and there are many debates about habitat depth of the organisms, therefore, according to our new structure, we have removed the habitat depth analysis of proxies. So this section is no more in the manuscript.

Author's changes in manuscript: L255-288 is removed from the manuscript.

Comment R2.17

-line 305: what do you mean by "Instead, the ensemble median (Fig. 2a) typically displays the common signal. In this case, it is the mean value of the fourth and fifth ensemble member out of eight models which are ordered according to ranked values"? Really not clear!

AC: As mentioned earlier in the manuscript, we have used eight PMIP3 models and due to space limitations, all individual model anomalies and their agreement/disagreement with the proxy-derived SST trends is not shown. Instead of that we have showed the ensemble median of them. First, we have ordered the models according to their ranked values. Then, we calculate the mean value of the fourth and fifth ensemble member out of eight models which represent the the ensemble median.

Author's changes in manuscript: Instead, the ensemble median of them is shown here (Fig. 2a) which typically displays the common signal.

Comment R2.18

- Discussion is too short, it must be clarified according to the objectives of the paper and the results (which also need to be more precise). A comparison between these

results and previous LGM simulations is required and must be discussed. A comparison with the CLIMAP values and other studies will also be welcome. A more in depth discussion of the agreement between terrestrial pollen-based temperature and model output is also needed as I have in mind that usually the models underestimates the values inferred from the data as it was the case in the previous PIMP simulations.

AC: Discussion is revised and mentioned at the Response letter to the Reviewer 1.

Author's changes in manuscript: See Response letter to the Reviewer 1 L302-09: The glacial ocean state has been under debate since the first reconstruction of the LGM sea surface temperatures and sea ice coverage by the CLIMAP project (CLIMAP Project Members, 1976). The SST reconstruction by the MARGO project (Waelbroeck et al., 2009) compared to CLIMAP indicates a more pronounced cooling in the eastern mid-latitude of the North Atlantic than in the western basin, a 1-3 oC cooling in the western Pacific warm pool (Fig. 1), as well as ice-free conditions in the Nordic Sea during glacial summer. According to MARGO study, in all the ocean basins, there is a large longitudinal gradient in LGM SST anomalies which are absent in the most of atmosphere-ocean coupled simulations of the PMIP2 project (Waelbroeck et al., 2009). A rather uniform SST cooling during the LGM in the range of 2-4 oC has been found (Fig. 1).

Comment R2.19

-in the 4.3 part, you discuss foraminifers, alkenones and MGCa ratio, but nothing is written about the dinos. I'm sure that a lot of papers are available. - Data model discrepancies can also been explained by the proxy itself or by the method (transfer function...); this point is important and need to be discussed - Line 410: You state that the proxy records used in most of the studies are more often located in coastal areas. I don't agree with you: Dino and forams records are not only located in coastal areas. - Figures : the order of each figure must be carefully checked in the text. The colors of the figures 5 a and 7a must be changed for more clarity.

AC: Same as comment R1.16.

[Figure]

Author's changes in manuscript: L360-412 is removed from the manuscript.

Please also note the supplement to this comment:
https://www.clim-past-discuss.net/cp-2018-9/cp-2018-9-AC3-supplement.pdf

———————————————————

[Figure]

**Fig. S3:** Reconstructed anomalies (in °C) between LGM and PI of mean temperature of the warmest month (MTWA) (a) and mean temperature of the coldest month (MTCO) (b) by Bartlein et al. (2011).

**Table 3:** Correlation and RMSE between COSMOS LIS and Bartlein et al. (2011) annual mean temperature (MAT), MTWA and MTCO.

|         | MAT         | MTWA        | MTCO        |
|---------|-------------|-------------|-------------|
|         | R, RMSE (‰) |             |             |
| LGMctl  | 0.40,5.30   | 0.50,6.55   | 0.46,7.14   |
| Gowan   | 0.29,5.38   | 0.44,6.31   | 0.43,7.05   |
| Ice6g   | 0.40,5.13   | 0.50,6.50   | 0.46,7.12   |
| Lambeck | 0.36,5.30   | 0.48,6.45   | 0.45,7.08   |
| Licc    | 0.30,5.33   | 0.46,6.48   | 0.44,7.40   |
| Tarasov | 0.41,5.08   | 0.49,6.62   | 0.45,7.15   |

Table S1: LIS reconstructions used in this study.

| LIS reconstructions | Name used in the study | Oceanic resolution | References |
|---------------------|------------------------|--------------------|------------|
| ICE-6G v2.0 | ICE-6G | 2°×1.2°, L31 | Argus and Peltier, 2010 |
| ANU | Lambeck | 1.4°×0.5°, L44 | Lambeck et al., 2014 |
| GLAC-1a | Tarasov | 1.25°×1°, L32 | Tarasov and Peltier, 2004 |
| Licciardi | Licc | 1°×1°, L60 | Licciardi et al., 1998 |
| ICESHEET 1.0 | Gowan | 1°×1°, L30 | Gowan et al., 2016 |
| PMIP3 LIS | LGMctl | 1°×0.5°, L51 | Braconnot et al., 2012 |

**Table S5:** Correlation and RMSE between PMIP3 models and Bartlein et al. (2011) annual mean temperature (MAT), MTWA and MTCO.

|           | MAT         | MTWA        | MTCO        |
|-----------|-------------|-------------|-------------|
|           | R, RMSE (‰) |             |             |
| IPSL-CM5A | 0.27,3.34   | 0.53,5.66   | 0.48,6.74   |
| MIROC-ESM | 0.25,5.11   | 0.53,6.03   | 0.45,8.10   |
| GISS-E2   | 0.21,8.58   | 0.08,8.46   | 0.25,9.42   |
| CCSM      | 0.25,4.76   | 0.48,5.78   | 0.41,7.74   |
| FGOALS-G2 | 0.15,4.04   | 0.53,5.65   | 0.44,6.89   |
| MRI-CGCM3 | 0.20,4.16   | 0.39,7.16   | 0.41,7.52   |
| CNRM-CM5  | 0.21,5.02   | 0.19,9.79   | 0.43,8.10   |
| MPI-ESM   | 0.23,4.29   | 0.50,5.99   | 0.43,7.65   |

**Fig. 1.**

[Figure]

**Figure 7:** Background color fill: simulated global pattern of annual mean surface temperature over land (T$_{2m}$) (in °C) changes between the eight PMIP3 model and PI climate. The circles localize the pollen-based reconstructed temperature changes by Bartlein et al. (2011).

**Fig. 2.**

---

## Author Comment (AC4) · 10 Aug 2018

Answer to reviewers' comments: A model-data comparison of the Last Glacial Maximum surface temperature changes

Akil Hossain, Xu Zhang, Gerrit Lohmann

Alfred Wegener Institute, Helmholtz Centre for Polar and Marine Research, Bremerhaven, Germany.

General remarks

We are very thankful to the editor and reviewers for the effort and time dedicated to the reviewing of our manuscript and for the helpful reviews. In order to address all

the concerns raised by the reviewers we have significantly restructured the manuscript and few new section. In this document, we supply detailed responses to all comments, suggestions and notes made by the two reviewers. We hope that the applied revisions are to the satisfaction of the reviewers and the editor.

Reviewer #1:

Comment R1.1

However, after reading the manuscript it is still unclear to me why and how the ice sheet reconstruction by Tarasov is better and what we have learned about marine temperature proxies apart from the known fact that they might be biased to variable seasons and depths. A paper that explicitly states 'comparison [of proxies] with outputs from climate model will help to understand the recording system itself' (L73) should deliver more and provide new insights, or directions, into how we can overcome the known recording biases. The approach taken by the authors is to simply look at what depth or season the marine proxy system correlates best. This implies that the recording bias may vary randomly from site to site. While there is nothing wrong with that approach as a starting point, we know that the ecology of the proxy carriers is not random (see e.g. the discussion section on alkenones or Leduc et al. [2010] or Jonkers and Kucera [2015]). The offsets between the annual mean SST and the reconstructed SST are thus likely to follow a systematic trend, likely with temperature. Rather than showing that ecology leaves an imprint on proxies (which is old news) the authors should investigate whether they see such trends in their comparison. A model that shows a pattern in the offset that is consistent with our understanding of the ecology of the recorder could arguably be considered to have more skill than one that doesn't. The opposite (no pattern, or random deviations) are more likely to be related to simple noise in the reconstructions or models. In this way models and data can be more meaningfully compared and new insights about the recording systems might be obtained.

Authors Comment (AC): Tarasov LIS reconstruction shows highest correlation and low-
est deviation with the land and marine proxies. Our submitted manuscript is examined the uncertainties in land and sea surface temperature of different ice sheet reconstructions and the PMIP3 models and we have compared the reconstructions of LGM temperature on land and in the ocean with climate models. We have also assess the potential recording biases in the proxy data and found that particularly in the marine realm there is considerable mismatch between the data and the models and these are due to seasonal recording biases in the proxies.

The offsets between the annual mean SST and the reconstructed SST are thus likely to follow a systematic trend, likely with temperature. Our model output partially agree with this pattern.

Considering comments from the reviewers we have changed the structure of our paper significantly and also we have improved the most part of the paper.

Comment R1.2

Related to this, it remains unclear how depth and season in the recording bias are separated? The same temperature can often be found at different times of the year or at different depths. How is this dealt with in paragraph 4.4? And what season is assumed in paragraph 4.3?

AC: Considering habitat depth of the planktonic organisms make our manuscript more complicated and there are many debates about habitat depth of the organisms, therefore, according to our new structure, we have removed the habitat depth analysis of proxies. So this section is no more in the manuscript.

Author's changes in manuscript: L255-288 is removed from the manuscript.

Comment R1.3

In addition, why is seasonal recording not considered for the terrestrial proxies? And is it right that the evaluation of the different ice sheet topographies in based solely on the terrestrial data? I couldn't find a figure or table with summary statistics.

AC: A subsection of seasonal parameters comparison also added to the manuscript.

Author's changes in manuscript:

L243: 3.1.5 Mean temperature of coldest and warmest month

According to Bartlein et al. (2011), July temperature in the northern hemisphere (southern hemisphere - December) has been combined with reconstructions of mean temperature of the warmest month (MTWA). Similarly, December temperature in the northern hemisphere (southern hemisphere - July) has been combined with reconstructions of mean temperature of the coldest month (MTCO) (Fig. S3, see also Bartlein et al., 2011).

During the LGM, Africa show warmer (1 to 4°C) than today in the reconstruction of MTWA (Fig. S3, see also Wu et al. 2007). A few sites in the northern hemisphere especially in the Alaska, reconstruction of warmer conditions as shown by seasonal temperature variable MTWA and similar or slightly warmer than today is registered chiefly in MTCO (Fig. S3) (Bartlein et al., 2011). The LIS was large enough to cause atmospheric circulation pattern reorganization. This reorganization could have originated in more southerly landward flow into Alaska, that would have produced advective warming in this region year-round (Bartlein et al., 2011). In general, the summer temperatures changes as represented by MTWA (Fig. S3) are smaller than the winter temperatures changes as represented by MTCO (Fig. S3, see also Bartlein et al., 2011).

For a comparison with proxy data, the warmest and coldest months of the model results have been compared with the seasonal temperature variables MTWA and MTCO. For MTWA, the highest correlation coefficient and lowest deviations are found for the LGMctl (R = 0.50, RMSE = 6.5‰ and Ice6g_LIS (R = 0.50, RMSE = 6.5‰ ice-sheet reconstruction and the lowest correlation coefficient and largest deviations for the Gowan_NAIS (R = 0.44, RMSE = 6.3‰ (Fig. 5). Similarly, for MTCO, the highest correlation coefficient and lowest deviations are also found for the LGMctl (R = 0.46)

[Figure]

and Ice6g_LIS (R = 0.46) and the lowest correlation coefficient for the Gowan_NAIS (R = 0.43) (Table 3). Overall, the correlation coefficient value for warmest and coldest months of the model has been increased than the model Annual mean value (Table 3).

L291: 3.2.2 Land Surface temperature changes

The annual mean SAT of PMIP3 LGM climate is on average 4.5 oC colder than the PI climate and CNRM is comparatively warmer (annual mean temperature -2.6 oC) than other models. PMIP3 model results have been compared with the LGM continental temperature reconstruction by Bartlein et al. (2011). The reconstructions show year-round cooling during the LGM over the continents except a few sites in Alaska (Fig. 7) (Bartlein et al., 2011). Similar as SST reconstructions, among the eight PMIP3 model, IPSL-CM5A-LR (R = 0.27, RMSE = 3.3‰ shows the highest correlation (Table S5), although most of the model show low correlation coefficient with the reconstructed data-set. MTWA (highest R is 0.53) show higher correlation than MAT and MTCO (highest R is 0.27 and 0.48). Overall, the correlation between model and data has been increased for MTWA and MTCO than the model Annual mean value (Table S5).

Comment R1.4

Equally importantly, the comparison between the reconstructions and the models could be improved. A simple correlation can be very misleading and the RMSE (deviation from the 1:1 line, why in per mille?) is a much more useful measure of the difference. Moreover, there is no statistical treatment of the uncertainties in the data or the model (at the minimum interannual variability in the model and the reported errors on the re-constructions should be taken into account). None of the statements about significance are accompanied by an explanation how this was determined and at what confidence level. This leaves the reader wondering whether the differences between the different ice sheet configurations or the different season/depth biases are real or meaningful. This is crucial as many differences between the models are very small. At some places in the manuscript the authors mention uncertainty in the models too. It would be good

if they discuss this more upfront. With so many models and different configurations of the same model (in this case the ice sheet topography) there are many degrees of freedom and there is a large chance of being right for the wrong reasons, not only because the proxies are biased (L163). How do the authors deal with that? Related to this, what have we learned about the model (configuration)? If some of the observed differences between the model runs are real/significant, then why? Where? Can the authors go deeper into the mechanisms or the physics that explain the differences?

AC: In our study, correlation coefficients between the reconstructions and the models show the similar pattern as RMSE value. As a unit of RMSE we have used per mille.

Discussion about potential uncertainties in the model is added to the manuscript.

Author's changes in manuscript:

Different local feedbacks working in upwelling systems might complicate the SST data-model comparison, since local cooling can occur within regions where widespread warming is found (Leduc et al., 2010b). Similarly, mismatches can be occurred due to difficulties in capturing variations in oceanic fronts in the climate models.

Figure 4b shows the difference between best-fit seasonal SST and temperature recorded by proxies. In the North Atlantic, still there is a big difference between the best-fit SST and temperature recorded by proxies especially for dinoflagellates (Fig. 4b). The observed mismatch between modelled and reconstructed LGM climate evolution is might be related to the lack of representativeness of long-term temperature anomalies in climate models.

The large discrepancy between data and model is likely caused by the large uncertainties in the reconstructed data as well as model deficiencies.

The interpretation of our data-model comparison suggests Mg/Ca proxies are winter biased, while foraminifera, dinoflagellates, and alkenones are summer biased. We find the similar results by using the COSMOS model LIS simulations and the PMIP3

simulations indicates that the deviation between model outputs and proxy data does not seem to be due to specific climate models, but because of a robust feature of LGM climate simulations with coupled climate models. One hypothesis is that proxies can therefore correctly capture local temperature trends that is not possible to simulate by the models. A possible way to test this effect is to use a new ocean model of high resolution with deep water formation areas up to 7 km and highly sensitive coastal areas to external forcing (Scholz et al., 2013) and apply this model to the LGM.

Palaeoclimate information collected from data-model comparisons are difficult to be put into a context which goes beyond a description of observed data-model discrepancies, as both proxy reconstructions and climate models are imperfect and have many different characteristics. Proxy reconstructions are patchy and sparse, and can be affected by different local processes and proxy specificities, which are not always counted in proxy reconstructions. Usually, palaeoclimatologists tend to collect data in the regions where the signal is clear and where sedimentation allows it. Therefore, there is a possibility of overestimation of the SST signals due to selection of the sites. Regional dynamics and spatially heterogenous patterns provide an additional uncertainty for our proxy data and model comparison.

For our model-data comparison, it is worth to mention that climate models have limitations in spatial resolution and are unable to represent the full complexity of the physical Earth System. The proxy records used in most of the studies are more often located in coastal areas, and climate models do not well represent these regions because of their low resolution (Lohmann et al., 2013). Coastal areas may be particularly sensitive to external forcing, as their thermal inertia is lower than the open ocean due to land-ocean interactions and a shallower thermocline. Moreover, the representation of mixed layer dynamics may be essential to improve climate simulations and its agreement with palaeoceanographic reconstructions.

Comment R1.5

In addition, the manuscript lacks a clear separation between results and discussion and the discussion section itself does hardly discuss the results, but rather summarises what others have said about potential recording biases in marine proxies. A lot of this could be placed in the introduction instead. Finally, there are numerous spelling and style errors. I have indicated some in the line-by-line comments below, but I recommend that the authors thoroughly proofread a revised version.

AC: The results and discussion parts are significantly restructured and edited. The manuscript is thoroughly checked and proofread for spelling and style errors.

Author's changes in manuscript:

3.1.4 Land Surface temperature changes

The annual mean SAT of the LGMctl run is 5.9 oC colder than the modelled PI climate. Most regions show a rather uniform cooling for all of the model runs in the range of −4 to −8 oC (Fig. 5). Alaska is the only region that is warmer than average in the model because of the increased distance to sea ice covered Arctic Ocean regions during the LGM, possibly due to the glacial sea level drop of approximately 120 m (Werner et al., 2016). The cold regions are mostly adjacent to the FIS and LIS, e.g., most of central North America and central Europe. There is another region of exceptional cooling located in northern Siberia where the temperature decreased down to −15 oC. The results agree with the temperature change of ensemble-mean LGM by the fully coupled climate simulations within the CMIP5/PMIP3 and PMIP2 projects (Braconnot et al., 2007; Harrison et al., 2014).

For a comparison with proxy data, the model results have been compared with the LGM continental temperature reconstruction by Bartlein et al. (2011), which is mainly based on plant macrofossil and subfossil pollen data. The highest correlation coefficient and lowest deviations are found for the Tarasov_LIS ice-sheet reconstruction (R = 0.41, RMSE = 5.0‰ and the lowest correlation coefficient and largest deviations for the Gowan_NAIS (R = 0.29, RMSE = 5.4‰ (Fig. 5, Table 3). Different core locations with the largest model-data variations are located near the boundary of the FIS and LIS. These deviations might simply be due to the coarse model resolution of 3.75o×3.75o that cannot resolve small-scale temperature changes close to the glacier area in sufficient detail. Overall, the model results agree well with the reconstructed LGM-PI temperature changes at the different core points (Fig. 5).

3.1.5 Mean temperature of coldest and warmest month

According to Bartlein et al. (2011), July temperature in the northern hemisphere (southern hemisphere - December) has been combined with reconstructions of mean temperature of the warmest month (MTWA). Similarly, December temperature in the northern hemisphere (southern hemisphere - July) has been combined with reconstructions of mean temperature of the coldest month (MTCO; Bartlein et al., 2011).

During the LGM, Africa show warmer (1 to 4°C) than today in the reconstruction of MTWA (Fig. S3, see also Wu et al. 2007). A few sites in the northern hemisphere especially in the Alaska, reconstruction of warmer conditions as shown by seasonal temperature variable MTWA and similar or slightly warmer than today is registered chiefly in MTCO (Fig. S3) (Bartlein et al., 2011). The LIS was large enough to cause atmospheric circulation pattern reorganization. This reorganization could have originated in more southerly landward flow into Alaska, that would have produced advective warming in this region year-round (Bartlein et al., 2011). In general, the summer temperatures changes as represented by MTWA are smaller than the winter temperatures changes as represented by MTCO (Fig. S3, see also Bartlein et al., 2011).

For a comparison with proxy data, the warmest and coldest months of the model results have been compared with the seasonal temperature variables MTWA and MTCO. For MTWA, the highest correlation coefficient and lowest deviations are found for the LGMctl (R = 0.50, RMSE = 6.5‰ and Ice6g_LIS (R = 0.50, RMSE = 6.5‰ ice-sheet reconstruction and the lowest correlation coefficient and largest deviations for the Gowan_NAIS (R = 0.44, RMSE = 6.3‰ (Fig. 5). Similarly, for MTCO, the highest

correlation coefficient and lowest deviations are also found for the LGMctl (R = 0.46) and Ice6g_LIS (R = 0.46) and the lowest correlation coefficient for the Gowan_NAIS (R = 0.43) (Table 3). Overall, the correlation coefficient value for warmest and coldest months of the model has been increased than the model Annual mean value (Table 3).

3.2 Data Model Comparison: PMIP3 models

3.2.1 Sea surface temperature changes

In most of the PMIP3 models, tropical cooling is more pronounced than in the MARGO reconstruction. The models and MARGO both show a more uniform LGM cooling in the Indian Ocean than in Pacific and Atlantic (Fig. 2, see also Wang et al., 2013). The greatest mismatch between the data and model is located in the North Atlantic and Northwestern Pacific. All of the models produced a significant cooling of 4-6 °C during LGM in the Northwestern Pacific, whereas a few MARGO records indicate that there was warming (2 °C or higher). The large discrepancy between data and model is likely caused by the large uncertainties in the reconstructed data as well as model deficiencies.

In this study, we analyze simulations from the PMIP3 model experiment to test the capability of current models to simulatie the LGM SSTs and land surface tempera-tures, with particular attention to model-data comparisons. Therefore, the anomaly of the LGM and PI simulated SST fields of all PMIP3 models have been compared with MARGO data-set and also with four individual proxy-based SSTs separately (Fig. 2, S4-S5). However, all of the considered PMIP3 models underestimate the temperature anomaly when compared to the proxy-inferred temperature data. A large mismatch and low correlation are found for most of the cases (listed in Table S3). Overall, the anomaly of the LGM and PI SST fields simulated by the PMIP3 models and the LIS simulation runs are comparable. Because of space limitations, all individual model anomalies and their agreement/disagreement with the proxy-derived SST trends are shown in the supplementary material (Figs. S4-5). Instead, the ensemble median of

them is shown here (Fig. 2a) which typically displays the common signal. In this case, it is the mean value of the fourth and fifth ensemble member out of eight models which are ordered according to ranked values. Among all models, IPSL-CM5A-LR shows the highest correlation and lowest RMSE with the MARGO data-set (Fig. 2b; Table S3). Since the results of the PMIP3 runs show large mismatches, we have compared with four MARGO proxies and seasonality. The seasonality in all models have been compared with individual proxies (listed in Table S4). In this case, correlation between PMIP3 models and proxies increases significantly. Overall, the agreement between the PMIP3 models and the SST reconstructions is similar to our COSMOS simulations.

3.2.2 Land Surface temperature changes

The annual mean land surface temperature of PMIP3 LGM climate is on average 4.5 oC colder than the PI climate and the CNRM-CM5 model is comparatively warmer (annual mean temperature -2.6 oC) than other models. PMIP3 model results have been compared with the LGM continental temperature reconstruction by Bartlein et al. (2011). The reconstructions show year-round cooling during the LGM over the continents except a few sites in Alaska (Fig. 7, see also Bartlein et al., 2011). Similar as SST reconstructions, among the eight PMIP3 model, IPSL-CM5A-LR (R = 0.27, RMSE = 3.3‰ shows the highest correlation (Table S5), although most of the model show low correlation coefficient with the annual mean reconstructed data-set. MTWA (highest R is 0.53) show higher correlation than MAT and MTCO (highest R is 0.27 and 0.48). Overall, the correlation between model and data has been increased for MTWA and MTCO than the model Annual mean value (Table S5).

4.2 Uncertainties of the land surface temperature reconstructions

From the analysis of the result show that, in general, changes in the land surface temperature in the model and proxy-inferred temperature data show a similar pattern and are in a good agreement although there is some mismatches at some cores location (Fig. 5). The simulated global-mean land surface temperature in LGM is 5.9 °C colder

than PI is comparable with the most recent estimate of the global-mean temperature anomalies based on reconstructions is 4.0 ± 0.8 °C (Annan and Hargreaves, 2013; Shakun et al., 2012), the global-mean cooling ranged from 3.6 to 5.7 °C in PMIP2 (Braconnot et al., 2007), as well as a global-mean cooling ranging from 4.41 to 5 °C in five PMIP3 models (Braconnot and Kageyama, 2015). It is also comparable with the LGM-PI simulation of CCSM3 revealed a global cooling of 4.5 °C with amplification of this cooling at high latitudes (Otto-Bliesner et al., 2006). Hence, the simulated estimate of this study appears reasonable, being slightly colder than the reconstructions and well within the range of previous simulations. Overall, the simulation of seasonal temperature over land is higher than seasonal temperature over the ocean (Annan and Hargreaves, 2015).

4.3 Seasonal biases

The interpretation of our data-model comparison suggests Mg/Ca proxies are winter biased, while foraminifera, dinoflagellates, and alkenones are summer biased. We find the similar results by using the COSMOS model LIS simulations and the PMIP3 simulations indicates that the deviation between model outputs and proxy data does not seem to be due to specific climate models, but because of a robust feature of LGM climate simulations with coupled climate models. One hypothesis is that proxies can therefore correctly capture local temperature trends that is not possible to simulate by the models. A possible way to test this effect is to use a new ocean model of high resolution with deep water formation areas up to 7 km and highly sensitive coastal areas to external forcing (Scholz et al., 2013) and apply this model to the LGM.

[revised manuscript text omitted]

The reconstructed LGM temperatures by dinocyst are much warmer than PI as well as much warmer than reconstructed by other proxies even after considering the best-fit SST (Fig. 3-4). One source of uncertainty in dinocyst proxies is low productivity and fluxes, particularly in the Nordic Sea, which could have resulted in over representation of transported material (de Vernal et al., 2005). The results from the seasonality are based on the model output which does not provide any diagnostic on the planktonic organisms real ecological behavior. However, they provide an oceanic regions map-
ping where even small changes in the ecology of planktonic organisms can have huge consequences on the reconstructed SST anomalies. It reinforces the idea that proxy organisms may be affected by ecological specificities (Leduc et al., 2010, Lohmann et al., 2013). Changes in recording season could have been caused by changes in insolation over the LGM or by related changes in the nutrient distribution and ocean temperature that the individual organisms are exposed to (Lohmann et al., 2013).

Palaeoclimate information collected from data-model comparisons are difficult to be put into a context which goes beyond a description of observed data-model discrepancies, as both proxy reconstructions and climate models are imperfect and have many different characteristics. Proxy reconstructions are patchy and sparse, and can be affected by different local processes and proxy specificities, which are not always counted in proxy reconstructions. Usually, palaeoclimatologists tend to collect data in the regions where the signal is clear and where sedimentation allows it. Therefore, there is a possibility of overestimation of the SST signals due to selection of the sites. Regional dynamics and spatially heterogenous patterns provide an additional uncertainty for our proxy data and model comparison.

For our model-data comparison, it is worth to mention that climate models have limitations in spatial resolution and are unable to represent the full complexity of the physical Earth System. The proxy records used in most of the studies are more often located in coastal areas, and climate models do not well represent these regions because of their low resolution (Lohmann et al., 2013). Coastal areas may be particularly sensitive to external forcing, as their thermal inertia is lower than the open ocean due to land-ocean interactions and a shallower thermocline. Moreover, the representation of mixed layer dynamics may be essential to improve climate simulations and its agreement with palaeoceanographic reconstructions.

Comment R1.6

Line by line comments L8: 'abrupt'. Reconsider wording What is meant here?

[Figure]

AC: Here, abrupt mean a large or steep change. The presence of vast Northern Hemisphere ice-sheets during the LGM caused a large changes in surface topography.

Author's changes in manuscript: No change.

Comment R1.7

L11-12: reword ' . . .pollen and plant macrofossil based. . .'

AC: This term has been revised. The annual temperature is mainly based on pollen data and sites with macrofossils data are very few for the LGM. That's why the term "plant macrofossils" is avoided.

Author's changes in manuscript: The term "plant macrofossils" is removed from the L11, L51, L110.

Comment R1.8

L16: it is the simulation using the Tarasov reconstruction that shows the highest correlation, not the reconstruction.

AC: This sentence has been revised.

Author's changes in manuscript: Among the six LIS reconstructions, simulation using Tarasov's LIS reconstruction shows the highest correlation with reconstructed terrestrial and SST.

Comment R1.9

L33: Project instead of Projection AC: This term has been corrected

Author's changes in manuscript: Paleoclimate Modeling Intercomparison Project (PMIP)

Comment R1.10

L40: please be more specific, uncertainty of what?

AC: Uncertainty of variables due to a large spread of reconstructed LIS with fundamental different geometries.

Author's changes in manuscript: uncertainty of variables

Comment R1.11

L54: please add a sentence or two to explain the link between the beginning and end of this paragraph. Importantly, Jonkers and Kucera [Jonkers and Kučera, 2017] –and before them several others [e.g. Mix, 1987; Schmidt, 1999; Schmidt and Mulitza, 2002; Skinner and Elderfield, 2005] – showed that there is predictability in the recording bias. This is an important point as it may help to distinguish between different models and or estimates of recording depth/season.

AC: This paragraph is revised and edited.

Author's changes in manuscript: A recent study by Jonkers and Kučera (2017) analyzed core top stable oxygen isotope ($\delta$18O) values of different planktonic foraminifera species. They found that planktonic foraminifera ecology exerts a significant influence on the proxy signal since bloom seasons of planktonic foraminifera vary at different locations and that there is predictability in the recording bias (Mix, 1987; Schmidt, 1999; Schmidt and Mulitza, 2002; Skinner and Elderfield, 2005; Jonkers and Kučera, 2017). Seasonality of planktonic foraminifera changes with temperature to minimize the environmental change that they experience. Habitat tracking can lead to reduce in the amplitude of this recorded environmental change and enable more improved reconstructions and data-model comparison (Jonkers and Kučera, 2017).

Comment R1.12

L74: replace 'will help' with 'might help'

AC: It has been replaced.

Author's changes in manuscript: Therefore, comparison with outputs from climate

model might help to understand the recording system itself.

Comment R1.13

L76: 'can force' – consider rewriting. Also, rewrite statement about all models in the next sentence. The PMIP3 ensemble does not contain all models of LGM climate.

AC: This portion has been revised in the manuscript.

Author's changes in manuscript: In this study, we have performed simulations with six LIS reconstructions in an atmosphere-ocean fully coupled climate model (COSMOS) (Zhang et al., 2013) to explore the "best-fit" LIS that might show a more consistent pattern with proxies during the LGM. In addition, proxy records are compared with eight PMIP3 model outputs.

Comment R1.14

L78: Strictly speaking there is no ecological effect on the proxy interpretation, there is an ecological effect on the recording of the climate sensor (proxy) [see for instance Evans et al., 2013].

AC: This portion has been revised in the manuscript.

Author's changes in manuscript: assess the potential ecological effect on the recording of the climate sensor (proxy).

Comment R1.15

L95: is Zhang et al. 2013 appropriate for the PMIP3 protocol? AC: Yes, Zhang et al. 2013 used external forcing and boundary conditions according to the PMIP3 protocol for the LGM. The respective boundary conditions for the LGM comprise greenhouse gas concentrations ($CO_2$ = 185 ppm; $CH_4$ =350 ppb; $N_2O$ = 200 ppb), orbital forcing, land surface topography, run-off routes, ocean bathymetry according to PMIP3 ice sheet reconstruction.

[Figure]

Author's changes in manuscript: No change

Comment R1.16

L109-134: what exactly is compared, the gridded products of the reconstructions or the individual sites? If the latter, why is the gridding explained and how were the data compared precisely?

AC: The individual sites of the reconstructions is compared with the model results but the gridding is described as an explanation of the dataset how it is organized. The individual sites of the temperature variables (annual mean temperature, MTWA and MTCO) of Bartlein et al. (2011) is compared with the LIS reconstructions and PMIP3 models. However, description of gridding is removed and paragraph is edited.

Author's changes in manuscript:

L109: The model results of our study is compared with the LGM continental temperature reconstruction by Bartlein et al. (2011), which is mainly based on subfossil pollen data. This dataset includes reconstructions of different temperature variables: mean temperature of the warmest month (MTWA), mean temperature of the coldest month (MTCO) and mean annual temperature (MAT) (Bartlein et al., 2011). The dataset considers a quantified estimate of combined uncertainties arising from the age scale uncertainties, data resolution and sampling, calibration model uncertainty, and analytical uncertainties (Bartlein et al., 2011). The individual sites of the temperature variables (annual mean temperature, MTWA and MTCO) of Bartlein et al. (2011) is compared with the LIS reconstructions of our model.

The Multiproxy Approach for the Reconstruction of the Glacial Ocean Surface (MARGO) project in 2009 has compiled and analyzed an updated synthesis of seasonal sea surface temperatures (SSTs) during the LGM (Kucera et al., 2005) based on all prevalent microfossil-based (planktonic foraminifera, diatoms, dinoflagellates and radiolarian abundances) and geochemical (alkenones and planktonic foraminifera

Mg/Ca) palaeothermometers from deep-sea sediments (Waelbroeck et al., 2009). Different types of records provide various information about ocean surface conditions: for example, alkenone data only give a measure of mean annual SST while foraminiferal assemblages can be analyzed statistically to obtain seasonal variation in SSTs (Waelbroeck et al., 2009). The MARGO dataset combines 696 individual SST reconstructions. The coverage is especially dense in the tropics, the North Atlantic and the Southern Ocean while several oceanic regions remain undersampled: for example, the subtropical gyres, especially in the Pacific Ocean (Waelbroeck et al., 2009).

Comment R1.17

L148: positions of brackets is incorrect.

AC: It has been corrected.

Author's changes in manuscript: found off adjacent to Greenland in the northern North Atlantic

Comment R1.18

L166-174: this is discussion. No references in results section.

AC This paragraph is moved to the discussion.

Author's changes in manuscript: New location of this paragraph is L302-09

Comment R1.19

L195: Change to 'Proxy-specific comparison' or equivalent.

AC: The section title has been revised.

Author's changes in manuscript: Proxy-specific comparison of SST annual mean

Comment R1.20

L231-240: discussion. It is also unclear to me what the main message of this paragraph

is.

AC: This section is moved to discussion and edited. It is discussed about previous research on seasonality and which agree with our results.

Author's changes in manuscript: L231-240 is moved to discussion.

Comment R1.21

L252: R = 0.01 means no correlation, not a positive one.

AC: This portion has been revised in the manuscript.

Author's changes in manuscript: Alkenones show positive correlation for the best-fit season (alkenones, R = 0.19) and dinoflagellates show no correlation

Comment R1.22

L256: the data is not composed of planktonic organisms, it's based on measurements of their fossil remains. Also reword 'shift in the different water columns'. L260: Coccolithophores (the alkenone-producing organisms) are phytoplankton and require light for photosynthesis. The same holds for other phytoplankton and symbiont-bearing planktonic foraminifera. 183 m seems rather deep for phytoplankton. I assume that light availability is not modelled, but the authors should look into this and assess whether the inferred recording depths (e.g. L269) are consistent with the ecology of the proxy carriers. There is also a lot of discussion in these sections. L270-274: this sentence begins and ends with different statements about the habitat depth of planktonic foraminifera. Please explain the difference, or discuss it. See also Rebotim et al. [2017] for a discussion on the variability of depth habitat.

AC: Considering habitat depth of the planktonic organisms make our manuscript more complicated and there are many debates about habitat depth of the organisms, therefore, according to our new structure, we have removed the habitat depth analysis of proxies. So this section is no more in the manuscript.

Author's changes in manuscript: L255-288 is removed from the manuscript.

Comment R1.23

L289-295: I disagree, if the data and the model disagree, and consistently disagree the reason is unlikely to be due to uncertainty in the data alone. Uncertainty in the data would lead to random variations around the mean value, not indicate consistent (temporal/spatial) changes. It is more likely that the mismatch is due to uncertainties/unknowns in both the data and the models. It would be good if the authors acknowledge that more.

AC: Yes, I agree with this comment, the disagreement between data and model is not uncertainty in the data alone. It might be caused by misinterpreted and/or biased proxy records as well as by model deficiencies. In our case, we have compared data with different PMIP3 models and observed that the relation we found between proxy-derived and modelled SSTs and land surface temperature is not model dependent. However, we have discuss about model deficiencies and uncertainties in the data in the discussion part.

Author's changes in manuscript: See answer to the comment R1.31

Comment R1.24

L327-329: this section on sediment traps needs referencing. It is also well known that there is no uniform seasonality of planktonic foraminifera, rather seasonality varies spatially [Jonkers and Kučera, 2015; Tolderlund and Bé, 1971] and has hence likely varied in the past.

AC: It from the same reference from the next sentences (Glacial Ocean Atlas, 2017). Yes, overall there is no uniform seasonality of planktonic foraminifera, rather seasonality varies spatially but in our case we found in the North Atlantic the best agreement of planktonic foraminifera for local summer.

Author's changes in manuscript: reference 'Glacial Ocean Atlas, 2017' is added for the

sediment trap.

Comment R1.25

L336-337: please be specific: uncertainty for planktonic foraminifera proxies, not the foraminifera themselves. Moreover, this not only holds for planktonic foraminifera, but for all proxy carriers with a short (< 1 year) life span [e.g. for coccolithophores that produce the alkenones Rosell-Melé and Prahl, 2013].

AC: Yes, it is uncertainty for planktonic foraminifera proxies.

Author's changes in manuscript: L331: planktonic foraminifera proxies.

Comment R1.26

L344-357: so it seems that there is a pattern in the season that is preferably reflected in the UK37 ratio. Is this resolved in the model-data mismatch? Does any model yield data more consistent with such a pattern? It is this kind of analysis that is lacking from the present manuscript.

AC: Yes, there is a pattern in the season that is preferably reflected in the UK37 ratio. In some part model output agree with that. Model agreement and disagreement is added to the manuscript.

Comment R1.27

L364: proxies are not exposed to nutrient conditions, the organisms are.

AC: It is corrected.

Author's changes in manuscript: Changes in recording season could have been caused by changes in insolation over the LGM or by related changes in the nutrient distribution and ocean temperature that the individual organisms are exposed to.

Comment R1.28

L377: Deuser and Ross and Anand et al used the same sediment trap time series for

their analysis, so this is only regionally constrained information. Crucially, one cannot infer living depth from sediment traps (perhaps the authors mean calcification depth). L380-384: this idea is hardly new, Emiliani [Emiliani, 1954; 1955] already touched on this. Please include. L395: There is also observational data that shows the dampening effect of changing habitat of the proxy carrier [Ganssen and Kroon, 2000; Jonkers and Kučera, 2017].

AC: Same as comments R1.22

Author's changes in manuscript: L360-412 is removed from the manuscript.

Comment R1.29

L391: it is unclear what is meant with 'in such a way'.

AC: It means in a way they would likely try to hold their preferred ecological conditions by changing their blooming seasons to mitigates the climate changes. However, It is edited.

Author's changes in manuscript: Proxy-recording organisms would likely try to hold their preferred ecological conditions by changing their blooming seasons in a way which mitigates the climate changes (Mix, 1987).

Comment R1.30

L400: why on the contrary, I don't understand the difference. And please explain why it is important to model foraminifera growth, rather than abundance. Note also that Fraile et al used many more variables than temperature alone [Fraile et al., 2008] (in fact, more than Lombard) and see Kretschmer et al [Kretschmer et al., 2017] for an update of this model.

AC: It is corrected. As previously discussed in the paragraph that planktonic organisms have several limiting factors such as temperature, nutrient, and light-availability. When those factors alter oppositely, the organisms try to change their living season without

modifying their basic ecological requirements. To explain such changes an ecosystem models can be used to reproduce the growth of planktonic foraminifera (Lombard et al., 2011) which also explain foraminifera abundance.

Comment R1.31

L406-412: I think a more upfront discussion of inherent uncertainties in the model is essential and should be placed not at the end of the discussion and include more than just model resolution.

AC: Discussion about potential uncertainties in the model is added in the earlier sections.

Author's changes in manuscript:

Different local feedbacks working in upwelling systems might complicate the SST data-model comparison, since local cooling can occur within regions where widespread warming is found (Leduc et al., 2010b). Similarly, mismatches can be occurred due to difficulties in capturing variations in oceanic fronts in the climate models.

Figure 4b shows the difference between best-fit seasonal SST and temperature recorded by proxies. In the North Atlantic, still there is a big difference between the best-fit SST and temperature recorded by proxies especially for dinoflagellates (Fig. 4b). The observed mismatch between modelled and reconstructed LGM climate evolution is might be related to the lack of representativeness of long-term temperature anomalies in climate models.

The large discrepancy between data and model is likely caused by the large uncertainties in the reconstructed data as well as model deficiencies.

The interpretation of our data-model comparison suggests Mg/Ca proxies are winter biased, while foraminifera, dinoflagellates, and alkenones are summer biased. We find the similar results by using the COSMOS model LIS simulations and the PMIP3 simulations indicates that the deviation between model outputs and proxy data does

not seem to be due to specific climate models, but because of a robust feature of LGM climate simulations with coupled climate models. One hypothesis is that proxies can therefore correctly capture local temperature trends that is not possible to simulate by the models. A possible way to test this effect is to use a new ocean model of high resolution with deep water formation areas up to 7 km and highly sensitive coastal areas to external forcing (Scholz et al., 2013) and apply this model to the LGM.

Palaeoclimate information collected from data-model comparisons are difficult to be put into a context which goes beyond a description of observed data-model discrepancies, as both proxy reconstructions and climate models are imperfect and have many different characteristics. Proxy reconstructions are patchy and sparse, and can be affected by different local processes and proxy specificities, which are not always counted in proxy reconstructions. Usually, palaeoclimatologists tend to collect data in the regions where the signal is clear and where sedimentation allows it. Therefore, there is a possibility of overestimation of the SST signals due to selection of the sites. Regional dynamics and spatially heterogenous patterns provide an additional uncertainty for our proxy data and model comparison.

For our model-data comparison, it is worth to mention that climate models have limitations in spatial resolution and are unable to represent the full complexity of the physical Earth System. The proxy records used in most of the studies are more often located in coastal areas, and climate models do not well represent these regions because of their low resolution (Lohmann et al., 2013). Coastal areas may be particularly sensitive to external forcing, as their thermal inertia is lower than the open ocean due to land-ocean interactions and a shallower thermocline. Moreover, the representation of mixed layer dynamics may be essential to improve climate simulations and its agreement with palaeoceanographic reconstructions.

Comment R1.32

L420-421: Sentence incomplete or wrong.

AC: Sentence is modified a little.

Author's changes in manuscript: It is assumed that the SST indicators have seasonal biases.

Comment R1.33

L423-427: this fundamental mismatch between the models and the data is mentioned here for the first time. It deserves mentioning in the results and discussion. As to the question whether it is the models or the data that cause this discrepancy, it is important to note that our current understanding of proxy carriers (in particular planktonic foraminifera) is that they tend to underestimate the environmental change (see suggested references and studies cited in the manuscript). Such homeostatic behaviour only exacerbates the mismatch.

AC: This comments is taken into account and a section of data model discrepancies is added to the discussion part.

Author's changes in manuscript: See answer to the comment R1.31

Comment R1.34

Fig. S1 is directly copied from the MARGO paper, I don't know if this is appropriate with regards to copy rights etc.

AC: We already have the permission from Nature Geoscience to reuse this figure.

Author's changes in manuscript: Fig. S1: Distribution of MARGO data points, indicating also which proxy was measured at each location (Waelbroeck et al., 2009 ©Nature Geoscience).

Comment R1.35

Table 1: why is there no RMSE for the Tarasov reconstruction? Also, none of the errors have units. Similarly, the legends in the figures often lack units.

AC: RMSE value for the Tarasov reconstruction has been added. Units for errors and legends in the figures and has been revised in the manuscript.

Author's changes in manuscript: RMSE value of Foraminifera is 2.65‰ MgCa is 5.90‰ Dinos is 6.64‰ and Uk37 is 3.44‰Units for error is added at the Figure 5 and Table 1-3, S3-S5. Units for legends is added to all the figures

Please also note the supplement to this comment:
https://www.clim-past-discuss.net/cp-2018-9/cp-2018-9-AC4-supplement.pdf

**LGMctl vs Foraminifera Best Fit Season**

**Fora-Seasonal Best Fit SST**

**LGMctl vs MgCa Best Fit Season**

**MgCa-Seasonal Best Fit SST**

**LGMctl vs Dinoflagellates Best Fit Season**

**Dinoflagellates-Seasonal Best Fit SST**

**LGMctl vs UK37 Best Fit Season**

**UK37-Seasonal Best Fit SST**

DJF   MAM   JJA   SON   Mean

0.5  1  1.5  2  3  4  5  6  7  8

a)

b)

Figure 4: (a) The circles localize the foraminifera, MgCa, dinoflagellates and $U^{k}_{37}$ records and the colors fill of the circles represent the seasonal/annual mean in which the reconstruction agrees best with model. (b) Colors fill of the circles show the anomalies between proxies and temperature trend (in °C) recorded by corresponding seasonal/annual mean shown in (a) at the sample locations.

**Fig. 1.**

[Figure]

**Figure 7:** Background color fill: simulated global pattern of annual mean surface temperature over land (T₂ₘ) (in °C) changes between the eight PMIP3 model and PI climate. The circles localize the pollen-based reconstructed temperature changes by Bartlein et al. (2011).

**Fig. 2.**

[Figure]

[Figure]

**Fig. S3:** Reconstructed anomalies (in °C) between LGM and PI of mean temperature of the warmest month (MTWA) (a) and mean temperature of the coldest month (MTCO) (b) by Bartlein et al. (2011).

**Table 3:** Correlation and RMSE between COSMOS LIS and Bartlein et al. (2011) annual mean temperature (MAT), MTWA and MTCO.

|         | MAT       | MTWA      | MTCO      |
|---------|-----------|-----------|-----------|
|         | R, RMSE (‰) |          |           |
| LGMctl  | 0.40,5.30 | 0.50,6.55 | 0.46,7.14 |
| Gowan   | 0.29,5.38 | 0.44,6.31 | 0.43,7.05 |
| Ice6g   | 0.40,5.13 | 0.50,6.50 | 0.46,7.12 |
| Lambeck | 0.36,5.30 | 0.48,6.45 | 0.45,7.08 |
| Licc    | 0.30,5.33 | 0.46,6.48 | 0.44,7.40 |
| Tarasov | 0.41,5.08 | 0.49,6.62 | 0.45,7.15 |

Table S1: LIS reconstructions used in this study.

| LIS reconstructions | Name used in the study | Oceanic resolution | References |
|---------------------|------------------------|--------------------|------------|
| ICE-6G v2.0         | ICE-6G                 | 2°×1.2°, L31       | Argus and Peltier, 2010 |
| ANU                 | Lambeck                | 1.4°×0.5°, L44     | Lambeck et al., 2014 |
| GLAC-1a             | Tarasov                | 1.25°×1°, L32      | Tarasov and Peltier, 2004 |
| Licciardi           | Licc                   | 1°×1°, L60         | Licciardi et al., 1998 |
| ICESHEET 1.0        | Gowan                  | 1°×1°, L30         | Gowan et al., 2016 |
| PMIP3 LIS           | LGMctl                 | 1°×0.5°, L51       | Braconnot et al., 2012 |

**Table S5:** Correlation and RMSE between PMIP3 models and Bartlein et al. (2011) annual mean temperature (MAT), MTWA and MTCO.

|           | MAT       | MTWA      | MTCO      |
|-----------|-----------|-----------|-----------|
|           | R, RMSE (‰) |          |           |
| IPSL-CM5A | 0.27,3.34 | 0.53,5.66 | 0.48,6.74 |
| MIROC-ESM | 0.25,5.11 | 0.53,6.03 | 0.45,8.10 |
| GISS-E2   | 0.21,8.58 | 0.08,8.46 | 0.25,9.42 |
| CCSM      | 0.25,4.76 | 0.48,5.78 | 0.41,7.74 |
| FGOALS-G2 | 0.15,4.04 | 0.53,5.65 | 0.44,6.89 |
| MRI-CGCM3 | 0.20,4.16 | 0.39,7.16 | 0.41,7.52 |
| CNRM-CM5  | 0.21,5.02 | 0.19,9.79 | 0.43,8.10 |
| MPI-ESM   | 0.23,4.29 | 0.50,5.99 | 0.43,7.65 |

**Fig. 3.**